# Sample from What You See: Visuomotor Policy Learning via Diffusion Bridge with Observation-Embedded Stochastic Differential Equation

Zhaoyang Liu [1 2]   Mokai Pan [1]   Zhongyi Wang [1]   Kaizhen Zhu [1]   Haotao Lu [1]   Haipeng Zhang [1]   Jingya Wang [1]   Ye Shi [1 2 *]

## Abstract

Imitation learning with diffusion models has advanced robotic control by capturing the multi-modal action distributions. However, existing methods typically treat observations only as high-level conditions to the denoising network, rather than integrating them into the stochastic dynamics of the diffusion process itself. As a result, the sampling is forced to begin from random noise, weakening the coupling between perception and control and often yielding suboptimal performance. We propose BridgePolicy, a generative visuomotor policy that directly integrates observations into the stochastic dynamics via a diffusion-bridge formulation. By constructing an observation-informed trajectory, BridgePolicy enables sampling to start from a rich and informative prior rather than random noise, substantially improving precision and reliability in control. A key difficulty is that diffusion bridge normally connects distributions of matched dimensionality, while robotic observations are heterogeneous and not naturally aligned with actions. To overcome this, we introduce a semantic aligner to unify the visual and state inputs and align the observations with action representations, making diffusion bridge applicable to heterogeneous robot data. Extensive experiments across 52 simulation tasks on three benchmarks and 5 real-world tasks demonstrate that BridgePolicy consistently outperforms state-of-the-art generative policies. Our code is available at https://jianghcsr.github.io/BridgePolicy_page/.

[1]ShanghaiTech University, Shanghai, China [2]InstAdapt. Correspondence to: Ye Shi <shiye@shanghaitech.edu.cn>.

*Proceedings of the $43^{rd}$ International Conference on Machine Learning*, Seoul, South Korea. PMLR 306, 2026. Copyright 2026 by the author(s).

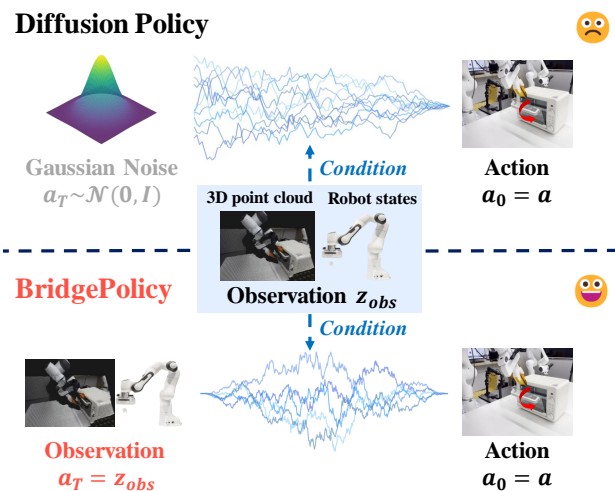

*Figure 1.* Comparison of Diffusion Policy and our BridgePolicy. The observation modeling way of BridgePolicy allows its sampling of BridgePolicy can start from an informative prior instead of the random noise in Diffusion Policy.

## 1. Introduction

Imitation learning (Osa et al., 2018) is a widely adopted learning paradigm in robotic learning (Li et al., 2024; Shafiullah et al., 2023; Ze et al., 2023b; Seo et al., 2023; Fu et al., 2024), where a robot is provided with a set of expert demonstrations and learns to mimic the provided demonstrations to perform the tasks effectively. Recently, generative models such as diffusion model (Ho et al., 2020; Song et al., 2020b; Dhariwal & Nichol, 2021; Ho & Salimans, 2022) and flow matching (Lipman et al., 2024; Liu et al., 2022; Albergo et al., 2023) gains its prominence owing to their capacity to capture multi-modal distributions and learn temporal dependency (Chi et al., 2023; Ze et al., 2023a; Zhang et al., 2025). These methods share a similar principle that perturb action chunks into random noise via a forward process defined by a Stochastic or Ordinary Differential Equation (SDE/ODE) and then train a neural network conditioned on observations to reverse this process, iteratively transforming noise into executable actions.

With this paradigm, Diffusion Policy (Chi et al., 2023) and

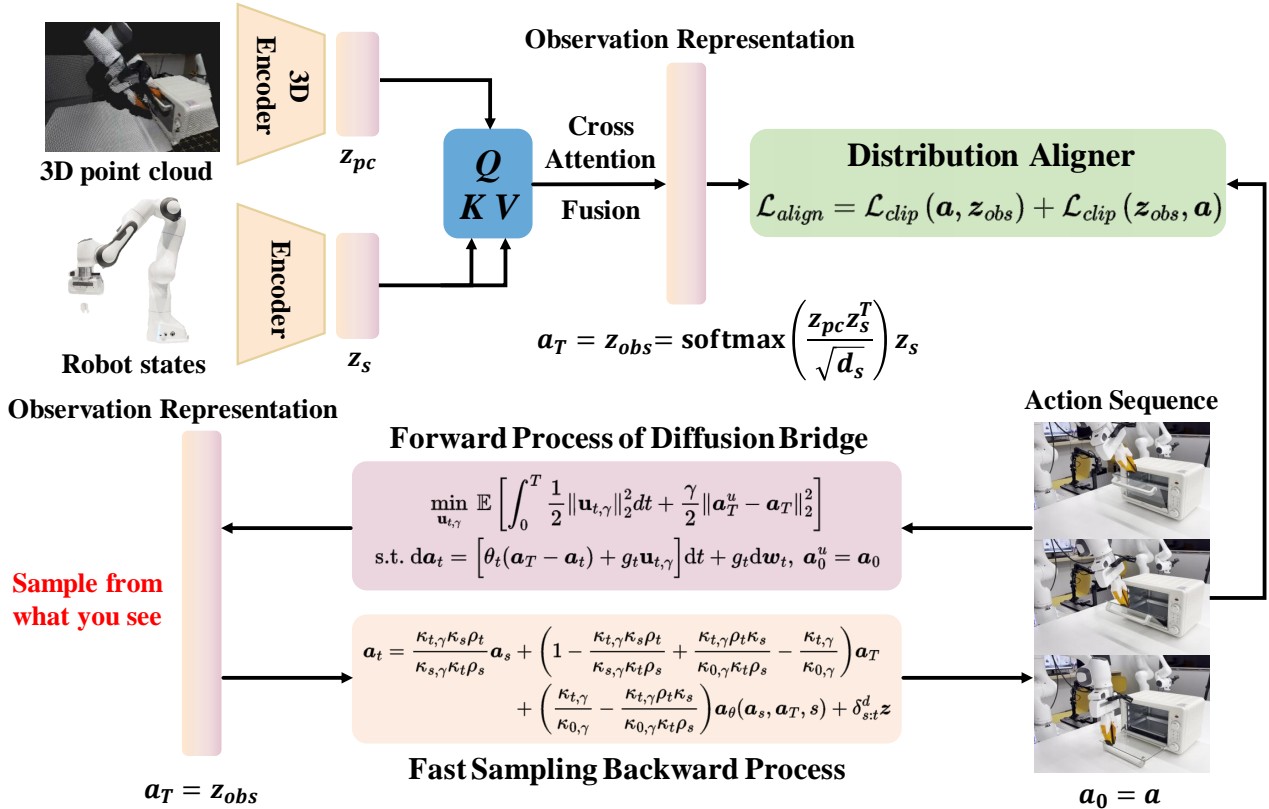

*Figure 2.* **Overview pipelines**. BridgePolicy explicitly embeds observations into the diffusion SDE trajectory via a diffusion-bridge formulation. The observation consists of robot states and point cloud. A semantic aligner address the challenges of heterogeneous distribution bridging and data shape mismatch, which enables effective exploitation of heterogeneous observations. During the inference, BridgePolicy samples from the observation and iteratively transforms it into the action through fast sampling.

3D Diffusion Policy (Ze et al., 2023a), known as DP and DP3, employ an SDE-defined forward process and train the neural network conditioned on visual inputs and robot states to control the denoising process to sample actions from the random noise. Similarly, FlowPolicy (Zhang et al., 2025) employs an ODE-defined forward and reverse process, which reduces the stochasticity during training and inference. Despite their success, current generative policies largely treat observations as high-level conditioning signals to the denoising network (Chi et al., 2023; Ze et al., 2023a; Zhang et al., 2025), rather than integrating them into the dynamics of the forward process. This underutilization forces sampling to begin from an uninformative random noise, weakening the coupling between perception and control and often yielding suboptimal performance.

Diffusion bridge has demonstrated considerable success in image restoration and translation (De Bortoli et al., 2021a; Yue et al., 2023; Li et al., 2023; Luo et al., 2023; Zhou et al., 2023), where it modifies the forward process such that the endpoint distribution naturally aligns with the desired conditioned distribution in standard diffusion. As shown in Figure 1, with this mathematically exact formulation of modeling

informative observations in the forward process rather than solely treating the observations as the external condition, the reverse process can start from the learned observation representations, a more informative prior instead of the uninformative random noise, thereby improving the precision of the generated actions. Building upon these insights, we propose that diffusion bridge can also serve as a more effective framework for visuomotor policy learning. Specifically, rather than adopting the conventional diffusion model, we formulate policy learning as the problem of learning a diffusion bridge, where observations are explicitly modeled in to the diffusion SDE trajectory itself through the framework of diffusion bridge (Zhu et al., 2025; Pan et al., 2025) rather than purely treating them as conditions, and the reverse process would be able to sample actions starting from a more informative prior of the observations instead of the uninformative random Gaussian noise. This deeper integration allows the policy to more effectively exploit observations, leading to more precise and reliable control.

However, formulating policy learning as the diffusion bridge brings two difficulties. First, robotic observations and actions are inherently heterogeneous, violating the standard

diffusion-bridge assumption that the connected endpoint distributions share the same dimensionality. Second, observations often include proprioceptive states, RGB-D vision, and language instructions, which do not admit a simple distribution mapping to the action space required by classical bridge formulations. Therefore, unifying the representation of heterogeneous observations and aligning the observation and action shapes constitute the critical challenges in making diffusion bridges applicable to policy learning. We introduce BridgePolicy, a generative visuomotor policy that directly samples actions from observation-informed priors rather than random noise. We construct a diffusion bridge that embeds observations within the diffusion SDE trajectory, enabling the model to fully exploit sensory inputs instead of using them solely as external conditions. To resolve the modality and shape mismatches, we design a semantic aligner that maps the observation representation into an action-aligned latent space, ensuring compatible endpoints for the bridge. Together, these modules allow the diffusion bridge to leverage heterogeneous data sources for policy learning. Our contributions are summarized as follows:

- We propose BridgePolicy, a novel generative visuomotor policy that explicitly embeds observations into the diffusion SDE trajectory via a diffusion-bridge formulation. This observation-informed SDE trajectory allows sampling to start from a meaningful prior rather than random noise, leading to more precise and reliable control.

- To make diffusion bridge applicable to heterogeneous robotic data, we introduce a semantic aligner that unify visual and state observations and align their representations with the action space. This resolves modality and shape mismatches, enabling effective use of heterogeneous observations for policy learning.

- Extensive experiments on 52 simulation tasks spanning three benchmarks and five real-world tasks show that BridgePolicy consistently outperforms prior generative policies, achieving state-of-the-art performance.

## 2. Related work

### 2.1. Generative Models in Robotic Learning

Diffusion models become state-of-the-art generative models, driven by their success in image generation (Song et al., 2020a; Rombach et al., 2022; Lipman et al., 2022). Due to the capacity of multi-modal and long-horizon modeling, they are widely used in reinforcement learning (Janner et al., 2022; Ding et al., 2024; 2025), imitation learning (Wu et al., 2024b; Pearce et al., 2023), and motion planning (Reuss et al., 2023; Sridhar et al., 2023; Xian & Gkanatsios, 2023; Prasad et al., 2024; Saha et al., 2024). In robotics, DP (Chi et al., 2023) and DP3 (Ze et al., 2024) adopt diffusion model

as the generative framework and condition the observations in the neural network to denoise the random noise into actions while flow based policies (Zhang et al., 2025; Hu et al., 2024) use the flow matching framework. Many other works aim to eliminate the dependence on random noise and generate actions directly from observations. BRIDGER (Chen et al., 2024) first trains a coarse policy and then adopts diffusion bridge to refine the actions, whose performance largely depends on the quality of the coarse policy. Another concurrent work, VITA (Gao et al., 2025), adopts a two-stage pipeline. It performs flow matching in a joint latent image-action space, and the executable actions must be recovered via a decoder. Our method models the heterogeneous observations in the diffusion SDE trajectory via diffusion bridge and generates the executable actions directly.

### 2.2. Diffusion Bridges for Generative Modeling

Diffusion bridges enable the transition between two arbitrary distributions without the need to initiate the diffusion process from the random noise, which provides a more principled and mathematically exact generation paradigm for distribution transformation. On one hand, Diffusion Schrödinger Bridges (Liu et al., 2023; Shi et al., 2024; De Bortoli et al., 2021b; Somnath et al., 2023; Chen et al., 2021; Deng et al., 2024; Wang et al., 2025) aim to determine a stochastic process by solving an entropy-regularized optimal transport problem between two distributions. However, its high computational complexity, particularly pronounced in high-dimensional settings or constraints, poses significant challenges for direct optimization. Other works (Zhou et al., 2023; Yue et al., 2023) incorporate Doob's $h$-transform into forward SDE, delivering remarkable performance on image restoration benchmarks. UniDB (Zhu et al., 2025) reformulates diffusion bridges through an SOC-based optimization problem, proving that Doob's $h$-transform is a special case of SOC theory, thereby unifying and generalizing existing $h$-transform-based diffusion bridge methods. In this work, we leverage the ability of diffusion bridge to connect heterogeneous distributions, specifically the heterogeneous observation distribution and the action distribution of an expert policy, and harness this mechanism for policy learning.

## 3. Policy Learning via Diffusion Bridge

### 3.1. BridgePolicy

Existing policy learning frameworks (Chi et al., 2023; Ze et al., 2024; Zhang et al., 2025; Hu et al., 2024) aim to learn a policy $\pi : O \to A$, which predict the actions $a \in A$ given the observations $o \in O$ which include 3D point cloud as the visual input and robot state and the actions are chunked into a short sequence of a trajectory finishing the task. Previous works such as DP3 (Ze et al., 2024) and FlowPolicy (Zhang

**Algorithm 1** BridgePolicy Training

**Input:** Actions $\boldsymbol{a}$, Observations $\boldsymbol{o} = \{\boldsymbol{o}_s, \boldsymbol{o}_{pc}\}$, loss weight $\alpha$, and gradient descent step size $\eta$.
**repeat**
    $\boldsymbol{z}_s = \mathbf{MLP}_{\phi_1}(\boldsymbol{o}_s)$, $\boldsymbol{z}_{pc} = \mathbf{MLP}_{\phi_2}(\boldsymbol{o}_{pc})$
    $\boldsymbol{z}_{obs} = \mathbf{softmax}(\boldsymbol{z}_{pc} \cdot \boldsymbol{z}_s^T / \sqrt{d_s})\boldsymbol{z}_s$
    $\boldsymbol{a}_0 = \boldsymbol{a}$, $\boldsymbol{a}_T = \boldsymbol{z}_{obs}$
    $t \sim \mathrm{Uniform}(\{1, ..., T\})$
    Compute $\mathcal{L} = \mathcal{L}_{DB} + \alpha\mathcal{L}_{align}$
    Update $(\theta, \phi_1, \phi_2) \leftarrow (\theta, \phi_1, \phi_2) - \eta\nabla\mathcal{L}$
**until** Converged

**Algorithm 2** BridgePolicy Inference

**Input:** Observations $\boldsymbol{o} = \{\boldsymbol{o}_s, \boldsymbol{o}_{pc}\}$, pre-trained data prediction model $\boldsymbol{a}_\theta$, and $M + 1$ time steps $\{t_i\}_{i=0}^M$ decreasing from $t_0 = T$ to $t_M = 0$.
$\boldsymbol{z}_s = \mathbf{MLP}_{\phi_1}(\boldsymbol{o}_s)$, $\boldsymbol{z}_{pc} = \mathbf{MLP}_{\phi_2}(\boldsymbol{o}_{pc})$
Initialize $\boldsymbol{a}_T = \boldsymbol{z}_{obs} = \mathbf{softmax}(\boldsymbol{z}_{pc} \cdot \boldsymbol{z}_s^T / \sqrt{d_s})\boldsymbol{z}_s$
**for** $i = 1$ **to** $M$ **do**
    Sample $\boldsymbol{\epsilon} \sim \mathcal{N}(0, I)$ if $i < M$, else $\boldsymbol{\epsilon} = 0$
    $\boldsymbol{a}_{t_i} \leftarrow \mathrm{Update}(\boldsymbol{a}_{t_{i-1}}, \boldsymbol{a}_T, \boldsymbol{a}_\theta, \boldsymbol{\epsilon}, t_{i-1})$ from (4)
**end for**
**Return** Actions $\boldsymbol{a}$

et al., 2025) formulate the policy as a conditional generative model, where actions are generated from random noises under the condition of observations $\boldsymbol{o}$ (Chi et al., 2023; Ze et al., 2024; Zhang et al., 2025). They purely treated the informative observations as conditional signals to guide the neural network during denoising.

**Forward Process of BridgePolicy.** Motivated by the recent advances of Diffusion Bridge (Zhou et al., 2023; Yue et al., 2023; Zhu et al., 2025), where conditions can be explicitly modeled in the forward diffusion SDE trajectory, we directly integrate the observations into the diffusion process to generate a more controllable and precise robot action. Here, our BridgePolicy adopt a unified diffusion bridge framework (Zhu et al., 2025), formulated under the Stochastic Optimal Control (SOC) theory, as the core learning paradigm. To be consistent with the notations of previous diffusion for visuomotor policy learning (Chi et al., 2023), we denote $\boldsymbol{a}_t$ as the interpolated robot action state at time $t$. Assuming that the action $\boldsymbol{a}$ and the observation $\boldsymbol{o}$ share the same dimension, we define the endpoints of our forward diffusion bridge process (2) as follows:

- The initial state $\boldsymbol{a}_0$: we set the initial state to be the action, i.e., $\boldsymbol{a}_0 = \boldsymbol{a}$.

- The terminal state $\boldsymbol{a}_T$: we set the terminal state to be the observation, i.e., $\boldsymbol{a}_T = \boldsymbol{o}$. It is a pivotal departure from standard diffusion-based or flow-based policy where $\boldsymbol{a}_T \sim \mathcal{N}(0, I)$ is typically the Gaussian noise.

According to UniDB (Zhu et al., 2025), we can construct the forward process of diffusion bridge as the following SOC optimization problem:

$$\min_{\mathbf{u}_{t,\gamma}} \mathbb{E}\left[\int_0^T \frac{1}{2}\|\mathbf{u}_{t,\gamma}\|_2^2 \, dt + \frac{\gamma}{2}\|\boldsymbol{a}_T^u - \boldsymbol{a}_T\|_2^2\right]$$
$$\text{s.t. } d\boldsymbol{a}_t = [\theta_t(\boldsymbol{a}_T - \boldsymbol{a}_t) + g_t\mathbf{u}_{t,\gamma}]dt + g_t d\boldsymbol{w}_t, \ \boldsymbol{a}_0^u = \boldsymbol{a}_0,$$
$$(1)$$

where $\boldsymbol{a}_0^u$ and $\boldsymbol{a}_T^u$ are the endpoints under control to be distinguished with the pre-defined endpoints, $\theta_t$ and $g_t$ are scalar-valued functions with the relation $g_t^2 = 2\lambda^2\theta_t$ and the steady variance level $\lambda^2$ is a given constant, $\boldsymbol{w}_t$ denotes the Wiener process, $\|\mathbf{u}_{t,\gamma}\|_2^2$ is the trajectory cost, and $\frac{\gamma}{2}\|\boldsymbol{a}_T^u - \boldsymbol{a}_T\|_2^2$ is the terminal cost with its penalty coefficient $\gamma$. The SOC problem (1) targets to design the optimal controller $\mathbf{u}_{t,\gamma}^*$ to drive the dynamic system from $\boldsymbol{a}_0$ to $\boldsymbol{a}_T$ with minimum cost. UniDB solved the problem (1) and obtained a closed-form optimally controlled forward SDE to transform $\boldsymbol{a}_0$ into $\boldsymbol{a}_T$ as:

$$d\boldsymbol{a}_t = \left(\theta_t + \frac{g_t^2 e^{-2\bar{\theta}_{t:T}}}{\gamma^{-1} + \bar{\sigma}_{t:T}^2}\right)(\boldsymbol{a}_T - \boldsymbol{a}_t)dt + g_t d\boldsymbol{w}_t, \quad (2)$$

where $\bar{\theta}_{s:t} = \int_s^t \theta_z dz$, $\bar{\theta}_t = \int_0^t \theta_z dz$, $\bar{\sigma}_{s:t}^2 = \lambda^2(1 - e^{-2\bar{\theta}_{s:t}})$, and $\bar{\sigma}_t^2 = \lambda^2(1 - e^{-2\bar{\theta}_t})$. Through the diffusion bridge paradigm, we successfully construct the mapping from actions $\boldsymbol{a}_0 = \boldsymbol{a}$ to observations $\boldsymbol{a}_T = \boldsymbol{o}$.

**Training Objective of BridgePolicy.** In order to learn such a bridge to reverse the process, we adopt the following reconstruction training objective to train a data prediction neural network $\boldsymbol{a}_\theta(\boldsymbol{a}_t, \boldsymbol{a}_T, t)$ that directly maps any $\boldsymbol{a}_t$ to the clean action $\boldsymbol{a}_0$:

$$\mathcal{L}_{DB} = \mathbb{E}_{t,\boldsymbol{a},\boldsymbol{a}_T,\boldsymbol{a}_t}\left[\|\boldsymbol{a}_\theta(\boldsymbol{a}_t, \boldsymbol{a}_T, t) - \boldsymbol{a}\|\right], \quad (3)$$

where we follow the $l_1$ norm in UniDB (Zhu et al., 2025) regarding the order of the norm, and provide the ablation study in Appendix. Consequently, BridgePolicy naturally integrates the informative observations $\boldsymbol{a}_T = \boldsymbol{o}$ both in the SDE trajectories and the neural networks and aims to learn a stochastic trajectory that transforms the observation $\boldsymbol{o}$ into a plausible action $\boldsymbol{a}$.

**Sampling Algorithm of BridgePolicy.** After we obtain the optimal model $\boldsymbol{a}_\theta^*(\boldsymbol{a}_t, \boldsymbol{a}_T, t)$, the decision making module of our BridgePolicy can be implemented as an iterative generation procedure starting from the given observation $\boldsymbol{a}_T = \boldsymbol{o}$. As for specially designed solvers, UniDB++ (Pan

et al., 2025) provides a training-free acceleration algorithm with data prediction $\boldsymbol{a}_\theta(\boldsymbol{a}_t, \boldsymbol{a}_T, t)$ that directly estimates the fixed and smooth target $\boldsymbol{a}_0$. With some notations $\rho_t = e^{\bar{\theta}_t}(1 - e^{-2\bar{\theta}_t})$, $\kappa_{t,\gamma} = e^{\bar{\theta}_{t:T}}((\gamma\lambda^2)^{-1} + 1 - e^{-2\bar{\theta}_{t:T}})$, $c_1 = (\gamma\lambda^2)^{-1}e^{2\bar{\theta}_T}$, $c_2 = e^{2\bar{\theta}_T} - 1$, $D = 2c_1c_2/(c_1 + c_2)^3$, $E = c_2^2/(c_1 + c_2)^2$, and $F = c_1^2/(c_1 + c_2)^2$ in UniDB++, the closed-form updating rule from time step $t$ to $s$ is formed as

$$\boldsymbol{a}_t = \frac{\kappa_{t,\gamma}\kappa_s\rho_t}{\kappa_{s,\gamma}\kappa_t\rho_s}\boldsymbol{a}_s + \left(\frac{\kappa_{t,\gamma}}{\kappa_{0,\gamma}} - \frac{\kappa_{t,\gamma}\rho_t\kappa_s}{\kappa_{0,\gamma}\kappa_t\rho_s}\right)\boldsymbol{a}_\theta(\boldsymbol{a}_s, \boldsymbol{a}_T, s)$$

$$+ \left(1 - \frac{\kappa_{t,\gamma}\kappa_s\rho_t}{\kappa_{s,\gamma}\kappa_t\rho_s} + \frac{\kappa_{t,\gamma}\rho_t\kappa_s}{\kappa_{0,\gamma}\kappa_t\rho_s} - \frac{\kappa_{t,\gamma}}{\kappa_{0,\gamma}}\right)\boldsymbol{a}_T + \delta_{s:t}^d\boldsymbol{\epsilon},$$

$$(\delta_{s:t}^d)^2 = \frac{\lambda^2\kappa_{t,\gamma}^2\rho_t^2}{\kappa_t^2}\left[E\frac{e^{2\bar{\theta}_s} - e^{2\bar{\theta}_t}}{(e^{2\bar{\theta}_t} - 1)(e^{2\bar{\theta}_s} - 1)}\right.$$

$$\left. + D\log\frac{\kappa_{s,\gamma}\rho_t}{\kappa_{t,\gamma}\rho_s} - F(\frac{e^{-\bar{\theta}_T - \bar{\theta}_t}}{\kappa_{t,\gamma}} - \frac{e^{-\bar{\theta}_T - \bar{\theta}_s}}{\kappa_{s,\gamma}})\right],$$

$$(4)$$

where $\boldsymbol{\epsilon} \sim \mathcal{N}(0, I)$ is the Guassian noise. Although the updating rule seems very complex on the coefficients of $\boldsymbol{a}_s$, $\boldsymbol{a}_T$, $\boldsymbol{a}_\theta$, and $\boldsymbol{\epsilon}$, actually these coefficients can be computed explicitly without incurring significant computational overhead, because they have closed forms and depend only on the specific time steps.

## 3.2. Modality Fusion and Alignment

While conceptually appealing under a uniform dimensionality assumption, directly applying the diffusion bridge paradigm remains infeasible, as such an assumption rarely holds in practice. This method still faces challenges of heterogeneous distribution bridging and data shape mismatch, which prevent the diffusion bridge from functioning effectively as a policy learning method. To address these limitations, we propose a novel framework that effectively bridges heterogeneous modalities through a carefully designed encoding and fusion pipeline.

**Encoding the Robot State and Point Cloud.** Our encoding pipeline begins by converting depth images into 3D point clouds, chosen for the efficiency. We then downsample them (512 or 1024 points in simulation; 2048 in real-world) using farthest point sampling (Qi et al., 2017). Finally, a lightweight MLP with LayerNorm and an MLP state encoder map the point cloud and robot state into a shared latent representation for fusion, which can be formulated as:

$$\boldsymbol{z}_s = \mathbf{MLP}_{\phi_1}(\boldsymbol{o}_s), \quad \boldsymbol{z}_{pc} = \mathbf{MLP}_{\phi_2}(\boldsymbol{o}_{pc}), \quad (5)$$

where $\phi_1$ and $\phi_2$ are the MLP network parameters, $\boldsymbol{o}_s$ is the robot state, and $\boldsymbol{o}_{pc}$ represents the point cloud.

**Multi-Modality Fusion.** To enable the diffusion bridge to map between heterogeneous modalities effectively, we introduce the multi-modality fusion module that integrates the

3D point cloud and robot state into a unified representation. This allows the policy to process diverse inputs coherently within our framework. Specifically, we employ the Cross-Attention mechanism (Rombach et al., 2022) to fuse the modalities:

$$\boldsymbol{a}_T := \boldsymbol{z}_{obs} = \mathbf{softmax}(\frac{\boldsymbol{z}_{pc}\boldsymbol{z}_s^T}{\sqrt{d_s}})\boldsymbol{z}_s, \quad (6)$$

where $\boldsymbol{z}_s$ represents the feature vector of the robot state, $d_s$ is the dimension of $\boldsymbol{z}_s$, and $\boldsymbol{z}_{pc}$ denotes the feature vector of the visual input. Here, $\boldsymbol{a}_T = \boldsymbol{z}_{obs}$ is the unified observation feature vector with the same shape of the action chunk, representing the unified observation representation.

We recognize that the terminal state $\boldsymbol{a}_T$ of diffusion bridge is an observation representation $\boldsymbol{z}_{obs}$ derived from the learned MLP and fusion via Eq. (5) and Eq. (6), which might introduce errors in the generated actions $\tilde{\boldsymbol{a}}_0$ if there exists errors during the encoder mapping. However, we demonstrate that the deviation in the generated action would not be too large, which leads to the following theorem:

**Theorem 3.1.** *Given the initial value $\boldsymbol{a}_T$ and Assumption B.1 on data prediction network $\boldsymbol{a}_\theta(\boldsymbol{a}_t, \boldsymbol{a}_T, t)$. Suppose $\tilde{\boldsymbol{a}}_T$ is the perturbed initial value from $\boldsymbol{a}_T$ and that the noise $\boldsymbol{\epsilon}$ during the sampling process is the same, then we have*

$$\|\tilde{\boldsymbol{a}}_0 - \boldsymbol{a}_0\| \le C\|\tilde{\boldsymbol{a}}_T - \boldsymbol{a}_T\| \quad (7)$$

*where $C > 0$ is a constant, $\boldsymbol{a}_0$ and $\tilde{\boldsymbol{a}}_0$ are respectively generated from $\boldsymbol{a}_T$ and $\tilde{\boldsymbol{a}}_T$ via the updating rule Eq. (4).*

Please refer to Appendix B for detailed proof. Theorem 3.1 indicates that the error of the action output can be linearly bounded by the error from the observation MLP. Since $C$ is difficult to determine analytically, we provide experimental results to empirically estimate the magnitude of $C$ in Appendix B. In our experiments, the magnitude of $C$ is found to be small, on the order of $10^{-2}$ to $10^{-3}$. Therefore, even when the observation MLP incurs fitting errors, the actions generated from the sampling algorithms of diffusion bridge would be constrained within a narrow range, preventing significant deviation.

**Modality Alignment.** Although multi-modality fusion aligns the observation and action distributions in shape, significant distributional differences still remain and the learned observation representation lacks sampling capability. To address this, we propose using the contrastive learning loss to train the aligner. The contrastive loss aligns the semantic proximity of the observation and action distributions. Specifically, we adopt CLIP loss (Radford et al., 2021) as the contrastive learning loss:

$$\mathcal{L}_{align} = \mathcal{L}_{clip}(\boldsymbol{a}, \boldsymbol{z}_{obs}) + \mathcal{L}_{clip}(\boldsymbol{z}_{obs}, \boldsymbol{a}),$$

$$\mathcal{L}_{clip}(\boldsymbol{x}, \boldsymbol{y}) = -\frac{1}{n}\sum_{j=1}^{n}\log\frac{\exp(x_j^\top y_j/\tau)}{\sum_{i=1}^{n}\exp(x_i^\top y_j/\tau)}, \quad (8)$$

*Table 1.* **Main Simulation Results.** Quantitative comparison on success rates among the baselines and our BridgePolicy. We evaluate 52 tasks across 3 benchmarks and report the average success rates of each benchmarks. For MetaWorld, we group the tasks based on their difficulty levels and report the average success rates.

| Methods\Task | MetaWorld Easy | MetaWorld Medium | MetaWorld Hard | MetaWorld Very Hard | DexArt | Adroit | Average |
|---|---|---|---|---|---|---|---|
| DP | $0.79_{\pm0.02}$ | $0.31_{\pm0.04}$ | $0.10_{\pm0.03}$ | $0.26_{\pm0.02}$ | $0.45_{\pm0.06}$ | $0.31_{\pm0.03}$ | 0.37 |
| DP3 | $0.87_{\pm0.01}$ | $0.61_{\pm0.03}$ | $0.40_{\pm0.05}$ | $0.51_{\pm0.02}$ | $0.57_{\pm0.06}$ | $0.68_{\pm0.04}$ | 0.60 |
| Simple DP3 | $0.86_{\pm0.02}$ | $0.59_{\pm0.04}$ | $0.38_{\pm0.04}$ | $0.47_{\pm0.03}$ | $0.48_{\pm0.05}$ | $0.68_{\pm0.02}$ | 0.58 |
| FlowPolicy | $0.86_{\pm0.02}$ | $0.67_{\pm0.02}$ | $\mathbf{0.59_{\pm0.02}}$ | $0.76_{\pm0.02}$ | $0.54_{\pm0.05}$ | $0.70_{\pm0.04}$ | 0.68 |
| VITA | $0.85_{\pm0.03}$ | $0.58_{\pm0.03}$ | $0.48_{\pm0.03}$ | $0.62_{\pm0.04}$ | $0.55_{\pm0.04}$ | $0.77_{\pm0.03}$ | 0.64 |
| **BridgePolicy (Ours)** | $\mathbf{0.91_{\pm0.03}}$ | $\mathbf{0.75_{\pm0.03}}$ | $0.58_{\pm0.02}$ | $\mathbf{0.79_{\pm0.02}}$ | $\mathbf{0.60_{\pm0.05}}$ | $\mathbf{0.81_{\pm0.04}}$ | **0.74** |

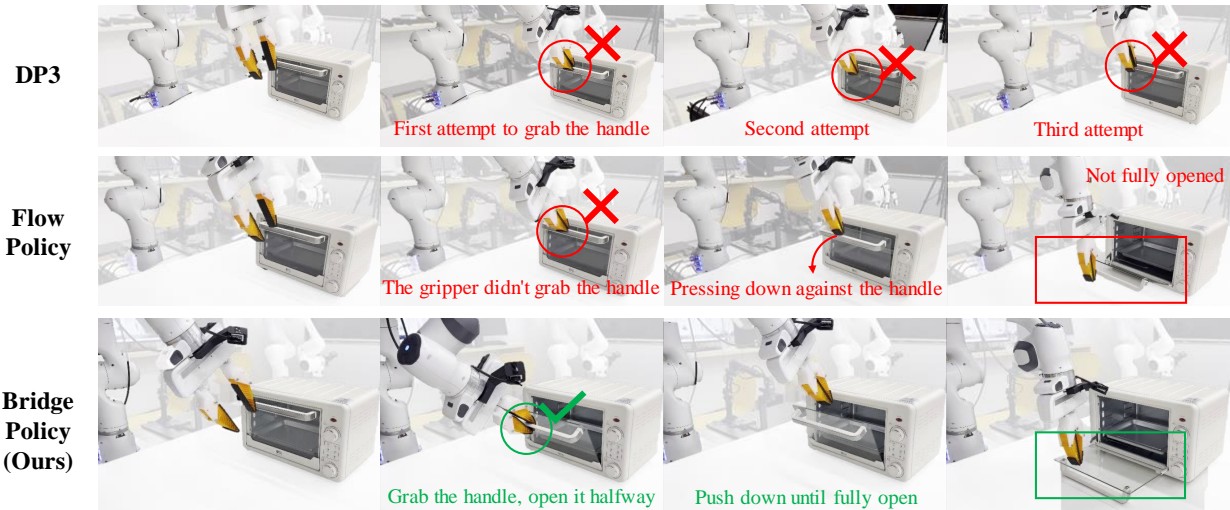

*Figure 3.* Real-robot comparative visualization of DP3, FlowPolicy, and BridgePolicy at four critical waypoints for Oven-Opening task.

where $n$ is the batch size and $\tau$ is the temperature parameter. The entire model involves the diffusion bridge and the modality fusion module and therefore, the overall training objective integrates the diffusion bridge loss (3) with the alignment loss (8):

$$\mathcal{L} = \mathcal{L}_{DB} + \alpha\mathcal{L}_{align}, \qquad (9)$$

where $\alpha$ is a positive weight. We provide the pseudo-code Algorithm 1 and Algorithm 2 for the training and inference process of our BridgePolicy. During the training phase, the observations are firstly fused and reshaped by the modality fusion module and aligner, and then alignment loss and diffusion bridge loss are computed by corresponding equations. After training, the observations are fused into the latent vector set as the starting point of the sampling process. This latent vector is subsequently updated iteratively by the designed solver (4), ultimately generating the actions.

## 4. Experiment

### 4.1. Simulation Experiment

**Simulation Benchmark.** In simulation tasks, we evaluate our BridgePolicy on 52 tasks across three benchmarks in-

cluding Adroit (Rajeswaran et al., 2017), DexArt (Bao et al., 2023), and MetaWorld (Yu et al., 2020). Adroit and DexArt focus on dexterous hand manipulation with varying complexity, while MetaWorld offers a broad spectrum of robotic arm tasks across different difficulty levels. Examples of tasks on these three benchmarks are illustrated in Figure 5.

**Expert Demonstration Collection.** We collect expert data with script policy in MetaWorld and adopt the VRL3 (Wang et al., 2022) and PPO (Schulman et al., 2017), two reinforcement learning (RL) algorithms, to collect expert data for Adroit and DexArt, respectively. We collect 10 episodes for tasks of the Adroit and the MetaWorld benchmarks and 100 episodes for the DexArt benchmark to train the policies.

**Baselines.** For comparison, we select state-of-the-art 2D-based methods DP (Chi et al., 2023) which takes the 2D image as its visual input and 3D conditional diffusion or flow based approaches including DP3 (Ze et al., 2024), simple DP3 (its lightweight version), FlowPolicy (Zhang et al., 2025) and VITA (Gao et al., 2025), which generates the action from random Gaussian noise and leverage 3D point cloud as the visual condition in the neural network, as baselines. We set the number of function evaluation (NFE) to 10

*Table 2.* **Main Real-world Results.** Quantitative comparison of success rates on real-world tasks among the baselines and BridgePolicy.

| Method | Oven-Closing | Oven-Opening | Pick Place | Pour | Unplug | Average |
|---|---|---|---|---|---|---|
| Simple DP3 | 0.8 | 0.6 | 0.6 | 0.6 | 0.7 | 0.66 |
| DP3 | 0.9 | 0.9 | 0.7 | 0.6 | 0.7 | 0.76 |
| FlowPolicy | **1.0** | 0.7 | 0.5 | 0.1 | 0.5 | 0.56 |
| BridgePolicy | **1.0** | **1.0** | **0.8** | **0.8** | **0.9** | **0.90** |

*Table 3.* Quantitative evaluation results (Success Rate) on different modality fusion methods (MetaWorld is denoted as MW in table).

| Task | Modality Fusion Method | |
|---|---|---|
| | Concatenation | Cross-Attention |
| Adroit Pen | 0.78 | **0.81** |
| Adroit Door | 0.59 | **0.665** |
| MW Handle-Pull | 0.55 | **0.63** |

*Table 4.* Quantitative evaluation results (Success Rate) on different Adroit tasks with different loss weights $\alpha$.

| Task | Loss weights $\alpha$ | | | | |
|---|---|---|---|---|---|
| | 0.0 | 0.5 | 1.0 | 2.0 | 5.0 |
| Adroit Door | 0.79 | 0.83 | **0.84** | **0.84** | 0.80 |
| DexArt Laptop | 0.85 | 0.86 | **0.91** | 0.86 | 0.89 |
| MW Coffee-Push | 0.91 | **0.99** | 0.91 | 0.91 | 0.92 |
| MW Pick-Place-Wall | 0.88 | 0.93 | **0.96** | 0.87 | 0.83 |

for all baselines except FlowPolicy. Based on the settings in the original FlowPolicy paper, we set its NFE to 1 instead of 10 to avoid the error accumulation issue that occurs in multi-step sampling of consistency flow matching (Sabour et al., 2025). For further details of the NFE setting, please refer to Appendix E.2.

**Evaluation and Implementation.** Followed by DP3 (Ze et al., 2024) and FlowPolicy (Zhang et al., 2025), we run 3 random seeds for each task. For each random seed, we train the models for 3000 epochs and evaluate the success rate on 20 episodes every 200 training epochs. We pick the five highest success rates per seed and report the mean success rate of the 3 random seeds. For fair comparison, we keep the model architecture consistent with the baselines, the amount of parameters comparable, and the shared hyperparameters, such as learning rates, weight decays of the optimizer, batch sizes, and training epochs. For each task, BridgePolicy is trained on a single NVIDIA RTX 4090 GPU. Please refer to Appendix D for further details of the implementation of BridgePolicy and other baselines.

**Performance in Simulator.** For Adroit and DexArt, we report the average success rates directly. Particularly for MetaWorld tasks, it is further categorized into four difficulty levels, with average rates reported for each. The quantitative results are reported in Table 1. Our proposed BridgePolicy demonstrates a consistent performance across all evaluated benchmarks, achieving the highest average success rates. All baseline methods show considerably lower success rates, particularly in more challenging settings (Hard and Very Hard in MetaWorld), which also showcases the robustness of our method in tasks with varying complexity. These results validate the effectiveness of BridgePolicy in leveraging the diffusion bridge formulation for improved policy learning in simulation environments. VITA's performance lies between DP3 and FlowPolicy, potentially because flow

matching is performed in a joint image–action latent space while executable actions are obtained with a decoder, which could inevitably introduce generalization error. Refer to Appendix F.5 for detailed success rates of individual tasks.

### 4.2. Real-world Experiment

**Experiment Setup and Evaluation.** We evaluate the BridgePolicy on 5 real-world tasks using the Franka Emika Panda robot. The point cloud is acquired using the ZED-2i camera. We evaluate the policies on 10 episodes of each task. The implementation remains the same as that in the experiments in simulation. We only increase the number of points of the point cloud to 2048 for a more dense representation of the real-world scenario. Examples of real-world tasks are illustrated in Figure 6. Real-world experiment environment setup details are provided in Appendix C.

**Expert Demonstration Collection.** The expert demonstrations are collected by the GELLO human teleoperation system (Wu et al., 2024a), manipulated by an experienced graduate. For each task, we collect 50 episodes to train the policy. Similarly, in the simulator and for a fair comparison, the expert demonstrations provided to all baseline methods and our BridgePolicy are identical. Refer to Appendix C for details of the real-world data collection.

**Quantitative Comparison on Real-World Task.** The success rates of real-world tasks are shown in Table 2. Under the identical constraint of training with the same dataset of 50 episodes, our BridgePolicy yields the highest success rate on all the tasks, with an average of 0.9, underscoring its effectiveness. Meanwhile, FlowPolicy exhibits a significant performance drop in the real world on tasks requiring location generalization, such as Pour and Pick-and-Place. DP3 is in the second place with a success rate of 0.78, and Simple DP3 only achieves a success rate of 0.65, possibly

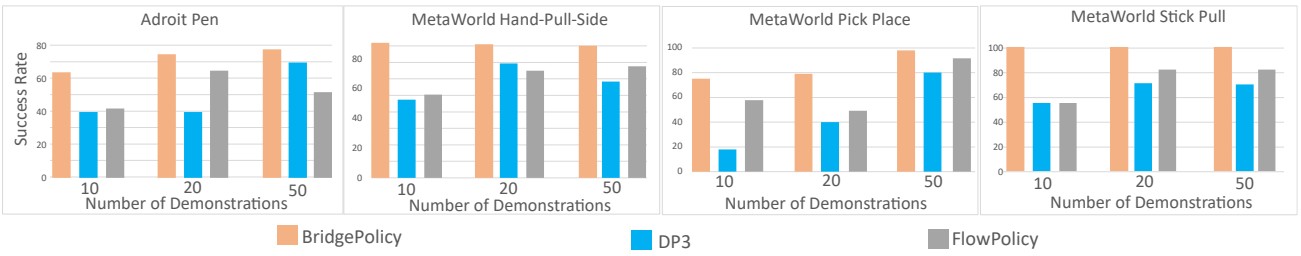

Figure 4. Ablation on the number of demonstrations. BridgePolicy shows its superiority with less training demonstrations.

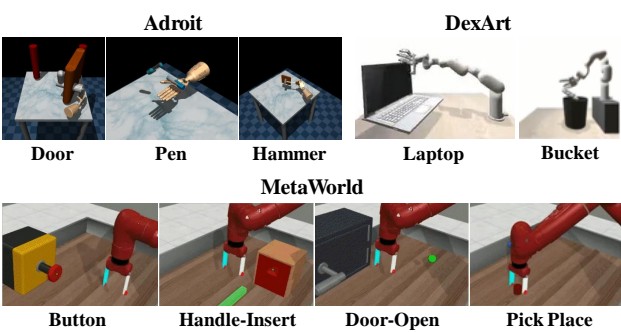

Figure 5. Examples of the simulation tasks.

due to the small number of parameters.

**Qualitive Comparison on Real-World Task.** Three real-world examples of Oven-Opening and Unplug generated by DP3, FlowPolicy, and BridgePolicy are shown in in Figure 3 and 14. The sequence presents the key frames of Franka finishing the task. The Oven-Opening task consists of 2 key stages: 1) grabbing the oven handle and opening it halfway; 2) moving the arm upon the half-opened door and pushing it down until it is fully opened. In the case of Oven-Opening, DP3 fails to finish the task while FlowPolicy and BridgePolicy succeed. BridgePolicy exactly finishes the 2 stages and fully opens the oven, while FlowPolicy fails to half-open the door in the first stage, and it presses against the handle directly, leaving the door not fully opened. We also observe similar results in the Unplug task, shown in Figure 14. DP3 and FlowPolicy extend their grippers towards the target location, but they do not precisely clamp the socket, thus fail to unplug successfully. BridgePolicy, on the other hand, successfully clamped the plug and pulled it out. Videos of real-world experiments are in Supplementary Materials.

### 4.3. Ablation Study

In this section, we analyze the performance of BridgePolicy under different hyperparameter settings. Additional ablation studies are provided in the Appendix F.

**Sample Efficiency.** The success rate of the agent in accomplishing tasks largely depends on the number of expert demonstrations. Here, we evaluate how the number of ex-

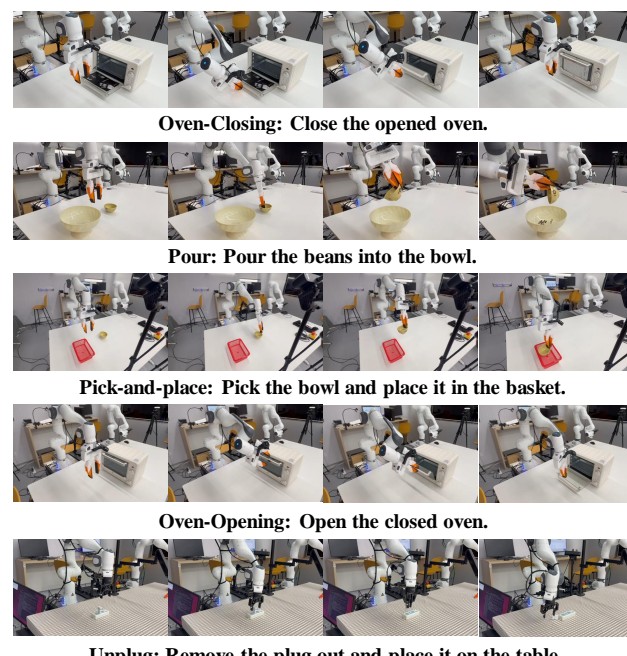

Oven-Closing: Close the opened oven.

Pour: Pour the beans into the bowl.

Pick-and-place: Pick the bowl and place it in the basket.

Oven-Opening: Open the closed oven.

Unplug: Remove the plug out and place it on the table.

Figure 6. Examples of the real-world tasks.

pert demonstrations would affect the success rate of the agent finishing the tasks. As shown in Figure 4, we select four tasks for this ablation study: the MetaWorld Hand-Pull-Side, Pick Place, Stick Pull and the Adroit Pen task. For all four tasks, increasing the number of expert demonstrations generally improves the agent's success rate. BridgePolicy shows the most stable and robust performance across all demonstration counts among the evaluated tasks, especially with fewer training samples. More sample efficiency ablations can be found in Appendix Section F.

**Parameter Sensitivity.** We demonstrate how the key parameters of the BridgePolicy would influence its performance. Specifically, we evaluate different weights $\alpha$ in the overall training loss (9) and the results are shown in Table 4. The results suggest that the distribution aligner can be trained better under a relatively small weight in policy learning. More ablation study results on the hyperparameters are in Appendix F.

**Modality Fusion.** We additionally explore the different

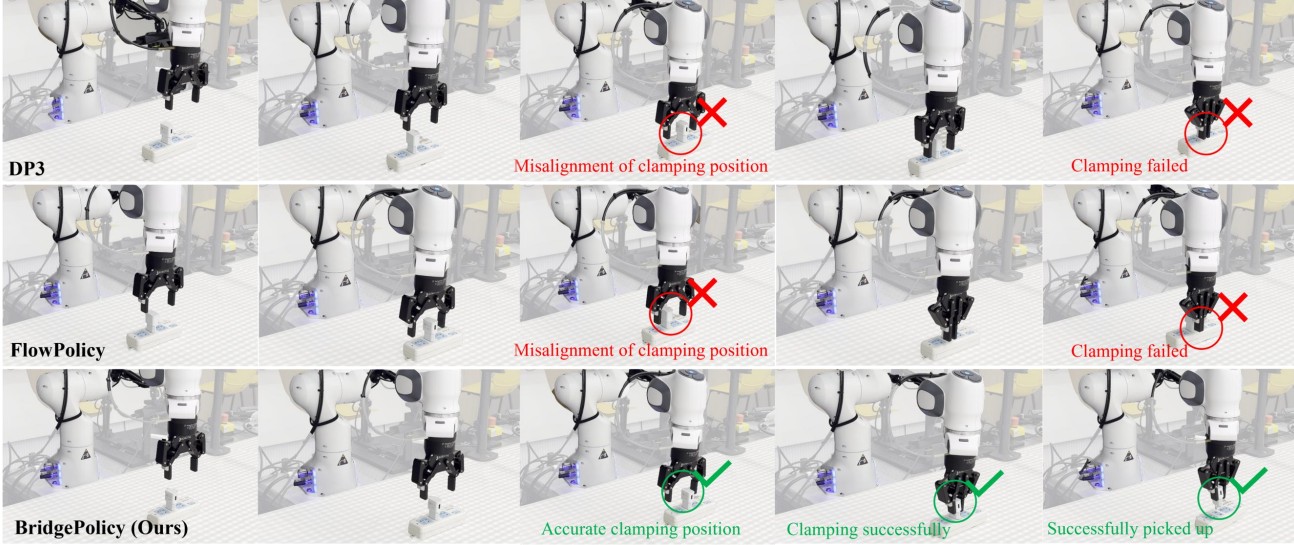

*Figure 7.* Real-robot comparative visualization of DP3, FlowPolicy, and BridgePolicy at four critical five points for Pick and Place task.

*Table 5.* Quantitative evaluation results (Success Rate) to evaluate the effectiveness of Diffusion Bridge in BridgePolicy.

| Task | Policy Head | | |
|---|---|---|---|
| | Regression | Diffusion | Bridge |
| Adroit Pen | 0.33 | 0.43 | **0.58** |
| Adroit Door | 0.00 | 0.68 | **0.84** |
| DexArt Laptop | 0.16 | 0.86 | **0.88** |
| DexArt Faucet | 0.00 | 0.37 | **0.39** |
| MW Basketball | 0.00 | 0.96 | **1.00** |
| MW Coffee Pull | 0.00 | 0.86 | **0.97** |

ways of fusing the different modalities of the observations. Modality fusion plays a crucial role in resolving the conflict between the multi-modal nature of the observations in policy learning and the distribution mapping of the diffusion bridge. We explore the impact of two modality fusion methods, Cross-Attention (Lin et al., 2022) and simple concatenation, on the performance. The results are shown in Table 3. When switching from Cross-Attention to simple concatenation, the performance consistently drops on all three evaluated tasks, which demonstrates the effectiveness of using Cross-Attention.

**Effectiveness of Diffusion Bridge.** We evaluate the effectiveness of Diffusion Bridge formulation compared to standard Diffusion and Regression. We remain the aligner and the modality fusion unchanged and use different policy heads to output actions. The results are shown in Table 5. Regression fails to behave effectively, often collapsing to near-zero success. Standard Diffusion significantly improves the performance over Regression. Diffusion Bridge gain the highest success rate, suggesting that its formulation provides a better-conditioned generation process and yields more precise actions.

## 5. Conclusion and Limitation

In this work, we present BridgePolicy, a novel generative policy learning paradigm that directly embeds the observations in the diffusion SDE trajectory via a diffusion bridge formulation rather than purely treating them as high-level conditions in the denoising network like conventional generative policies. This observation-informed SDE enables the sampling to start from a rich and meaningful prior, allowing the policy to more effectively exploit observations, leading to more precise and reliable control. BridgePolicy not only outperform the SOTA generative policies, but also extend the application of diffusion bridge to multi-modal and heterogeneous distribution matching. Extensive experiments on simulation and real-world tasks demonstrate its superiority over the current generative policy. Though BridgePolicy outperforms the SOTA generative policies, its sampling remains constrained by its SDE formulation, which limits the possibility of applying distillation methods that would enable one-step generation.

## Acknowledgment

This work was supported by the National Natural Science Foundation of China (62303319, 62406195), Shanghai Local College Capacity Building Program (23010503100), ShanghaiTech AI4S Initiative SHTAI4S202404, HPC Platform of ShanghaiTech University, and MoE Key Laboratory of Intelligent Perception and Human-Machine Collaboration (ShanghaiTech University), Shanghai Engineering Research Center of Intelligent Vision and Imaging. This work was also supported in part by computational resources provided by Fcloud CO., LTD.

## Impact Statement

This paper presents work whose goal is to advance the field of machine learning. There are many potential societal consequences of our work, none of which we feel must be specifically highlighted here.

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

## A. Overview

We provide additional information, results, and visualization to supplement the main paper. This Appendix are organized as follows:

- The proof of Theorem 3.1 and its empirical evaluation;

- Information of experimental setup and real-world data collection;

- Implementation details of BridgePolicy and Baselines;

- Training and inference details;

- Additional and expanded experimental results.

## B. Proof of Theorem 3.1 and the Empirical Evaluation

To prove Theorem 3.1, we make the following common assumption about the Lipschitz property of data prediction neural network $a_\theta$ (Lu et al., 2022; 2025):

**Assumption B.1.** The function $a_\theta(a_t, a_T, t)$ is Lipschitz w.r.t. to its parameters $a_t$ and $a_T$, i.e., there exists $L_1, L_2 > 0$ s.t. the following inequality holds,

$$\|a_\theta(a'_t, a''_T, t) - a_\theta(a_t, a_T, t)\| \le L_1 \|a'_t - a_t\| + L_2 \|a''_T - a_T\|. \tag{10}$$

**Theorem 3.1.** *Given the initial value $a_T$ and Assumption B.1 on data prediction network $a_\theta(a_t, a_T, t)$. Suppose $\tilde{a}_T$ is the perturbed initial value from $a_T$ and that the noise $\epsilon$ during the sampling process is the same, then we have*

$$\|\tilde{a}_0 - a_0\| \le C \|\tilde{a}_T - a_T\| \tag{7}$$

*where $C > 0$ is a constant, $a_0$ and $\tilde{a}_0$ are respectively generated from $a_T$ and $\tilde{a}_T$ via the updating rule Eq. (4).*

*Proof.* Consider the two updating rules from time step $s$ to $t \in [0, s]$:

$$
\begin{aligned}
a_t &= c_{1,t:s} a_s + c_{2,t:s} a_T + c_{3,t:s} a_\theta(a_s, a_T, s) + \delta^d_{t:s} \epsilon, \\
\tilde{a}_t &= c_{1,t:s} \tilde{a}_s + c_{2,t:s} \tilde{a}_T + c_{3,t:s} a_\theta(\tilde{a}_s, \tilde{a}_T, s) + \delta^d_{t:s} \epsilon,
\end{aligned}
\tag{11}
$$

where we simply denote the three coefficients w.r.t. $a_s$, $a_T$, and $a_\theta$ as $c_{1,t:s}$, $c_{2,t:s}$, and $c_{3,t:s}$, respectively, and the noise $\epsilon$. Define the error at time $t$ as

$$e_t \triangleq \|\tilde{a}_t - a_t\|. \tag{12}$$

Then we have

$$
\begin{aligned}
e_t &= \|\tilde{a}_t - a_t\| \\
&= \|c_{1,t:s}(\tilde{a}_s - a_s) + c_{2,t:s}(\tilde{a}_T - a_T) + c_{3,t:s}[a_\theta(\tilde{a}_s, \tilde{a}_T, s) - a_\theta(a_s, a_T, s)]\| \\
&\le^{(i)} c_{1,t:s}\|\tilde{a}_s - a_s\| + c_{2,t:s}\|\tilde{a}_T - a_T\| + c_{3,t:s}\|a_\theta(\tilde{a}_s, \tilde{a}_T, s) - a_\theta(a_s, a_T, s)\| \\
&\le^{(ii)} c_{1,t:s} e_s + c_{2,t:s} e_T + c_{3,t:s}(L_1 e_s + L_2 e_T) \\
&= (c_{1,t:s} + c_{3,t:s} L_1) e_s + (c_{2,t:s} + c_{3,t:s} L_2) e_T \\
&= C_{1,t:s} e_s + C_{2,t:s} e_T,
\end{aligned}
\tag{13}
$$

where $(i)$ follows from the triangle inequality, $(ii)$ follows from Eq. (10), and we simply denote $C_{1,t:s} \triangleq c_{1,t:s} + c_{3,t:s} L_1$ and $C_{2,t:s} \triangleq c_{2,t:s} + c_{3,t:s} L_2$. Given $M + 1$ time steps $\{t_i\}_{i=0}^M$ decreasing from $t_0 = T$ to $t_M = 0$, repeating the argument

(13), we have

$$
\begin{aligned}
\boldsymbol{e}_0 &\leq C_{1,t_M:t_{M-1}}\boldsymbol{e}_{t_{M-1}} + C_{2,t_M:t_{M-1}}\boldsymbol{e}_T \\
&\leq C_{1,t_M:t_{M-1}}C_{1,t_{M-1}:t_{M-2}}\boldsymbol{e}_{t_{M-2}} + (C_{1,t_M:t_{M-1}}C_{2,t_{M-1}:t_{M-2}} + C_{2,t_M:t_{M-1}})\boldsymbol{e}_T \\
&\leq \cdots \\
&\leq \prod_{i=1}^{M}C_{1,t_{M-i+1}:t_{M-i}}\boldsymbol{e}_{t_0} + \sum_{i=1}^{M}C_{2,t_{M-i+1}:t_{M-i}}\prod_{k=i+1}^{M}C_{1,t_{M-k+1}:t_{M-k}}\boldsymbol{e}_T \\
&= \left(\prod_{i=1}^{M}C_{1,t_{M-i+1}:t_{M-i}} + \sum_{i=1}^{M}C_{2,t_{M-i+1}:t_{M-i}}\prod_{k=i+1}^{M}C_{1,t_{M-k+1}:t_{M-k}}\right)\boldsymbol{e}_T.
\end{aligned}
\tag{14}
$$

Hence,

$$
\begin{aligned}
\boldsymbol{e}_0 &\leq C\boldsymbol{e}_T \\
\Leftrightarrow \quad \|\tilde{\boldsymbol{a}}_0 - \boldsymbol{a}_0\| &\leq C\|\tilde{\boldsymbol{a}}_T - \boldsymbol{a}_T\|,
\end{aligned}
\tag{15}
$$

where $C \triangleq \prod_{i=1}^{M}C_{1,t_{M-i+1}:t_{M-i}} + \sum_{i=1}^{M}C_{2,t_{M-i+1}:t_{M-i}}\prod_{k=i+1}^{M}C_{1,t_{M-k+1}:t_{M-k}}$ is a constant only depending on the finite number of sampling steps and Lipschitz constants in Assumption B.1, which concludes the proof.

$\square$

**Empirical Estimation of Constant Parameter $C$ in Theorem 3.1.** We conduct experiments on six tasks to empirically evaluate the magnitude of $C$. Specifically, we add different level of random perturbations to the same learned observation representation $\boldsymbol{a}_T = \boldsymbol{z}_{obs}$ and generate actions through the reverse process, after which we calculate the corresponding $C$. Here $M = 10$ is consistent with NFE in the simulation experiments. For each task, we average the results on 1000 sampling trials.

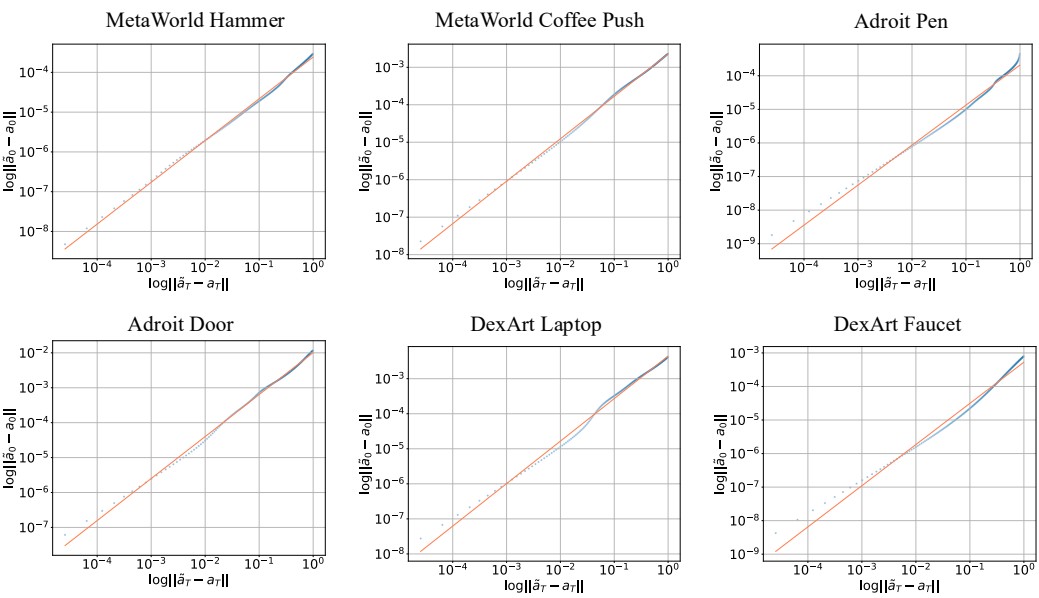

*Figure 8.* Empirical Evaluation of Constant $C$ in Theorem 3.1.

According to Theorem 3.1, after taking the logarithm of both sides of the inequality ($\log\|\tilde{\boldsymbol{a}}_0 - \boldsymbol{a}_0\| \leq \log C + \log\|\tilde{\boldsymbol{a}}_T - \boldsymbol{a}_T\|$), the corresponding error curve should lie below a straight line with a slope of approximately 1. Our experimental results are consistent with the statement of Theorem 3.1. In Figure 8, all fitted lines exhibit slopes roughly in the range of 1.09–1.23

and the constant $C$ falls in $10^{-2}$ and $10^{-3}$, which aligns with our formulation. The errors introduced by the observation MLP would not influence the generated action $a_0$ too much.

## C. Experimental Setup and Real-World Data Collection Details

### C.1. The Simulation and Real-World Task Example

The examples of the simulation and real-world tasks are shown in Figure 9.

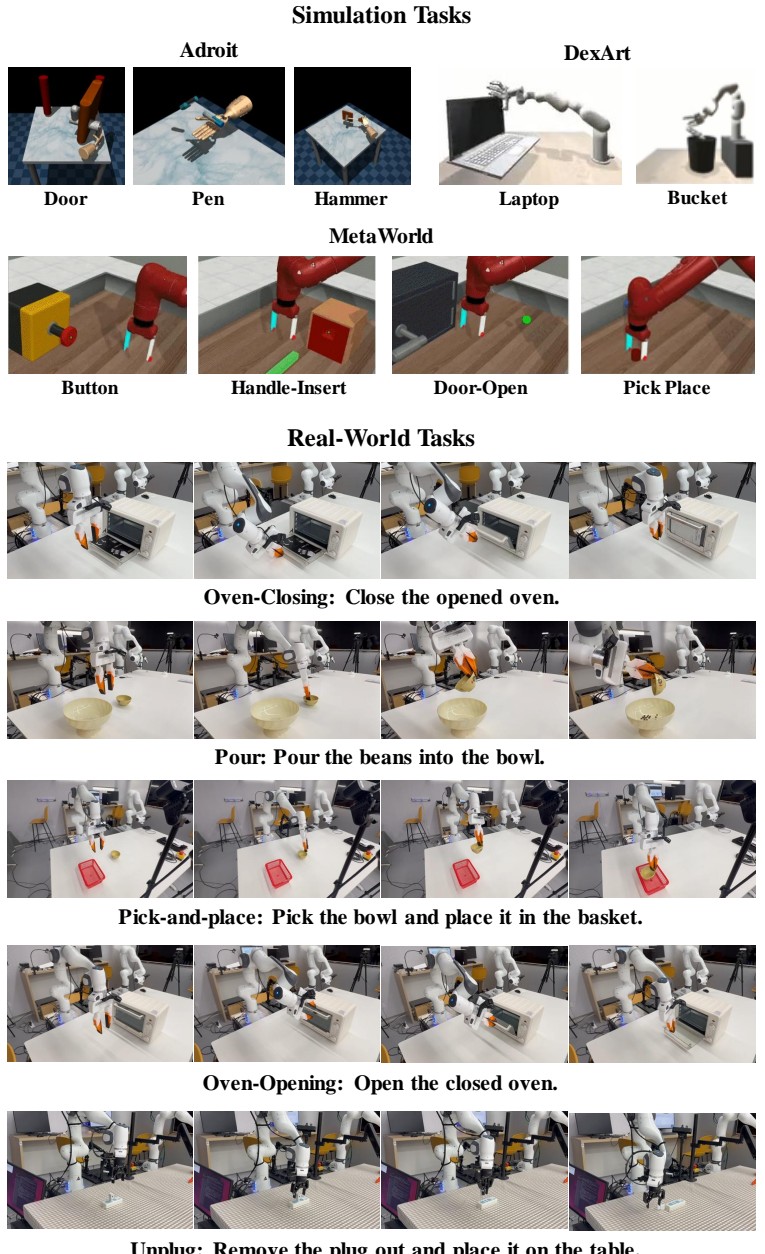

*Figure 9.* Examples of the simulation and real-world tasks.

### C.2. Real-World Experiment Setup

The real-world experiment was conducted on a Franka arm. We equipped it with two types of grippers, one from Robotiq and the other from FastUMI. The specific scenario setup is shown in Figure 12. The FastUMI gripper is used to finish the Pour, Oven-Opening, Oven-closing and Pick-and-Place tasks and the Robotiq gripper is used to finish the Unplug task.

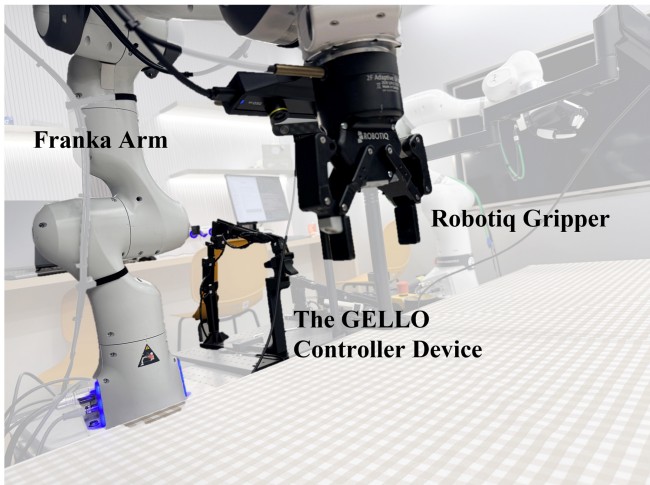

*Figure 10.* The GELLO human teleoperate system for real-world data collection.

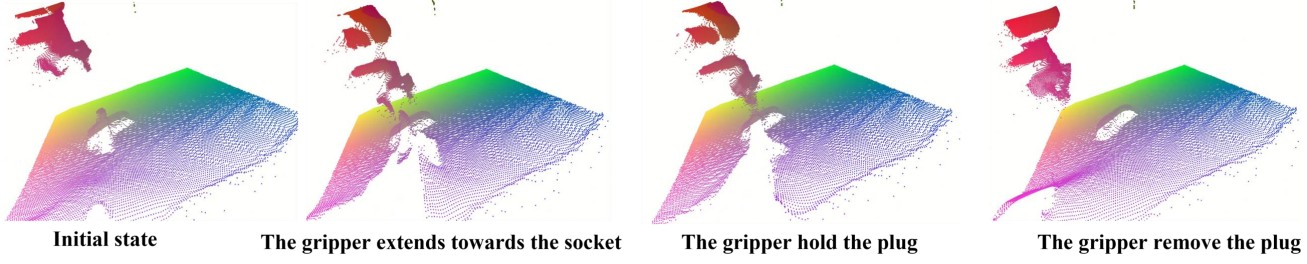

| Initial state | The gripper extends towards the socket | The gripper hold the plug | The gripper remove the plug |

*Figure 11.* The visualization of point cloud of the key frames of the Unplug task.

### C.3. Data Collection

We use the GELLO human teleoperate system (Wu et al., 2024a) to collect the data for real-world experiments. The GELLO controller device and the GELLO data collection system we use is illustrated in Figure 10. We perform hand-eye calibration in the real-world data collection. The calibration process enabled us to precisely determine the transformation relation from the camera coordinate system to the robot-base coordinate system. The observations including 3D point clouds and robot states and actions are recorded and consist the expert demonstration dataset. We use 3D point clouds as the robot's visual input, which is directly acquired by the ZED-2i camera, a binocular camera can generate the point cloud of the scene. An example of the point cloud acquired by the camera is shown in Figure 11. A single point cloud output from the ZED-2i camera contains 252,672 points. The point cloud input to the policies including BridePolicy and baselines was further downsampled to 2048 points using the Farthest Point Sampling (FPS) algorithm and done by the fpsample library (Han et al., 2023) for further acceleration to support the real-time inference.

## D. Model Implementation of BridgePolicy and Baselines

### D.1. Implementation of The Denoising Network

Following the implementation of the denoising model in DP3 (Ze et al., 2024) and FlowPolicy (Zhang et al., 2025), we adopt the U-Net architecture as the backbone of the denoising model in BridgePolicy and the baselines. The numbers of parameters in the denoising model in BridgePolicy and the baselines are the same which is approximately 255M. The

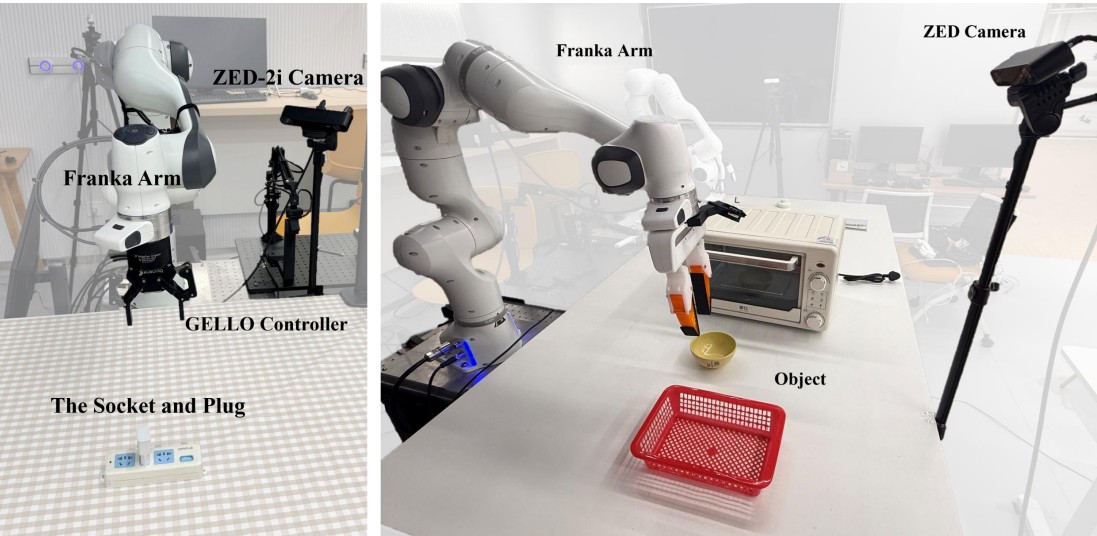

*Figure 12.* The experimental setup for the unplug task with Robotiq and FastUMI gripper.

*Table 6.* Evaluation of FlowPolicy using 10 step and 1 step for sampling.

|  | Flow (1 step) | Flow (10 step) |
|---|---|---|
| Adroit Hammer | 1.00 | 0.91 |
| Adroit Door | 0.53 | 0.50 |
| DexArt Faucet | 0.38 | 0.37 |
| DexArt Bucket | 0.32 | 0.26 |
| MetaWorld Hammer | 0.98 | 0.94 |
| MetaWorld Coffee-Pull | 0.96 | 0.89 |
| MetaWorld Soccer | 0.33 | 0.24 |
| Average | 0.64 | 0.59 |

BridgePolicy possess additional modules, the semantic aligner and the modality fusion module, which only have about less than 1M parameters.

## E. Training and Inference Details of BridgePolicy and Baselines

### E.1. Training

For the configurations such as the horizon length, the number of observation steps and the predicted action steps, we set them to the default setting in their publicly available code base and the corresponding configurations of BridgePolicy is aligned with DP3. For training of the aligner in BridgePolicy, we frozen the parameters after 50 epochs. Specifically, we list the hyperparameters used in BridgePolicy in Table 7 and 8.

### E.2. Inference

**Number of Sampling Steps.** We set the number of sampling steps to be 10 for BridgePolicy and baselines except the number of sampling steps of FlowPolicy is set to 1 according to the original paper (Zhang et al., 2025). We also conduct additional experiments to explore how the performance would change if we increase the number of sampling steps and the results are shown in Table 6. These results show that the performance degrades, possibly due to the error accumulation issue in consistency flow matching (Sabour et al., 2025).

*Table 7.* Hyperparameter setting of BridgePolicy in experiments (real-world).

| Hyperparameter | Value |
| --- | --- |
| Num epochs | 4000 |
| Batch Size | 256 |
| Horizon | 8 |
| Observation Steps | 2 |
| Action Steps | 4 |
| Num points | 2048 |
| Num train timesteps | 100 |
| Num inference steps | 10 |
| Learning Rate (LR) | 1.0e-4 |
| Aligner weight $\alpha$ | 1.0 |
| Weight decay | 1.0e-6 |

*Table 8.* Hyperparameter setting of BridgePolicy in experiments (simulation).

| Hyperparameter | Value |
| --- | --- |
| Num epochs | 3000 |
| Batch Size | 256 |
| Horizon | 8 |
| Observation Steps | 2 |
| Action Steps | 4 |
| Num points | 512/1024 |
| Num train timesteps | 100 |
| Num inference steps | 10 |
| Learning Rate (LR) | 1.0e-4 |
| Aligner weight $\alpha$ | 1.0 |
| Weight decay | 1.0e-6 |

# F. Additional Experimental Results

Here we provide additional experimental results.

## F.1. Additional Visualization of Real-World Tasks

Figure 13, 14, and 7 demonstrate the additional visualization of the real-world tasks between our BridgePolicy and baselines.

## F.2. More Ablation Results on $\alpha$ parameter

We provide more ablation results on the $\alpha$ parameter shown in Table 9 and 10. For most of the evaluated tasks, a smaller $\alpha$ less than 1 is better for training the policy.

*Table 9.* Ablation Study: Success Rates of Different $\alpha$ on MetaWorld

| $\alpha$ | MetaWorld | | | Avg |
| --- | --- | --- | --- | --- |
| | coffee-push | hammer | pick-place-wall | |
| 0.0 | 0.91 | 0.96 | 0.88 | 0.91 |
| 0.5 | 0.99 | 0.97 | 0.93 | 0.96 |
| 1.0 | 0.91 | 0.99 | 0.96 | 0.95 |
| 2.0 | 0.91 | 0.97 | 0.87 | 0.91 |
| 5.0 | 0.92 | 0.93 | 0.83 | 0.89 |

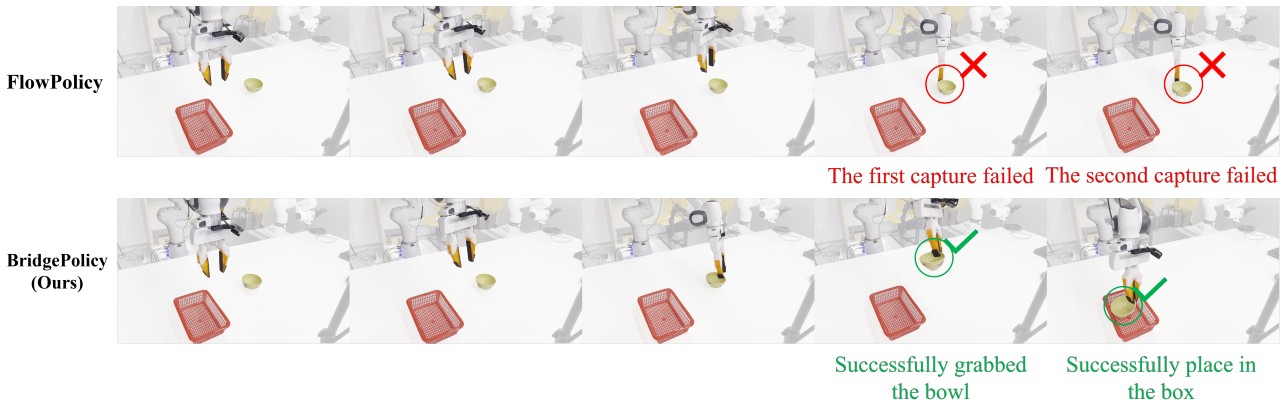

*Figure 13.* Real-robot comparative visualization of DP3, FlowPolicy, and BridgePolicy at five critical points for Pick and Place task.

### F.3. Parameter Sensitivity on $\gamma$ and $\lambda$

We demonstrate how the key parameters of the BridgePolicy would influence its performance. Specifically, we evaluate the combinations of two hyperparameters (the steady variance level $\lambda^2$ (over 255) and the penalty coefficient $\gamma$) in the UniDB (Zhu et al., 2025) framework and the results are shown in Table 11. Results in Table 11 demonstrates that $\lambda = 50$ and $\gamma = 10^7$ are the relatively better hyperparameter combination of the UniDB framework in policy learning.

### F.4. Ablation Results on Training with $L_1$ and $L_2$ norm objective

We validate the effect of using different training objective to train diffusion bridge and report the success rate of BridgePolicy. The results are shown in Table 12.

### F.5. Success Rates of Individual Tasks

We list the success rate of individual task of each benchmark in Table 13. The mean and standard deviation are computed over three random seeds.

### F.6. More Sample Efficiency

We provide additional sample efficiency ablation results in this section shown in Figure 15.

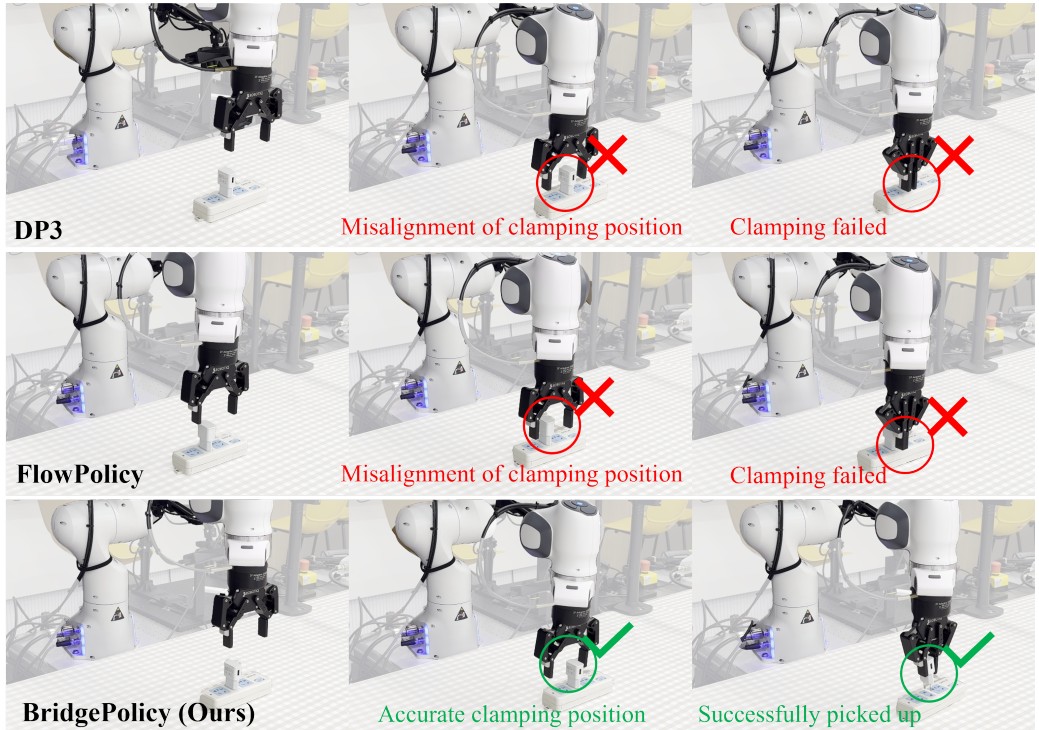

*Figure 14.* Real-robot comparative visualization of DP3, FlowPolicy, and BridgePolicy at four critical waypoints for Unplug task. BridgePolicy successfully grabbed the plug, while DP3 and FlowPolicy can only move to the approximate location.

*Table 10.* Ablation Study: Success Rates of Different $\alpha$ on Adroit and DexArt.

| $\alpha$ | Adroit | | | Adroit Avg | DexArt | | | | DexArt Avg |
|---|---|---|---|---|---|---|---|---|---|
| | hammer | door | pen | | laptop | faucet | toilet | bucket | |
| 0.0 | **1.00** | 0.79 | 0.56 | 0.78 | 0.85 | 0.42 | 0.74 | 0.34 | 0.58 |
| 0.5 | **1.00** | 0.83 | 0.51 | 0.78 | 0.86 | 0.41 | 0.75 | **0.35** | 0.59 |
| 1.0 | **1.00** | **0.84** | 0.54 | 0.79 | **0.91** | 0.44 | **0.76** | 0.30 | **0.60** |
| 2.0 | **1.00** | **0.84** | **0.62** | **0.82** | 0.86 | 0.42 | **0.76** | 0.32 | 0.59 |
| 5.0 | 0.97 | 0.80 | 0.57 | 0.78 | 0.89 | **0.43** | 0.73 | 0.30 | 0.58 |

*Table 11.* Quantitative evaluation results on different Adroit tasks with different steady variance levels $\lambda^2$ and penalty coefficients $\gamma$.

| Task | Hyperparameters $(\lambda, \gamma)$ | | | | | |
|---|---|---|---|---|---|---|
| | $(30, 10^7)$ | $(50, 10^7)$ | $(70, 10^7)$ | $(30, 10^5)$ | $(50, 10^5)$ | $(70, 10^5)$ |
| Adroit Hammer | 0.9 | **1.0** | **1.0** | 0.85 | **1.0** | **1.0** |
| Adroit Door | 0.75 | **0.85** | **0.85** | 0.65 | **0.85** | 0.825 |
| Adroit Pen | 0.65 | **0.75** | 0.65 | 0.65 | 0.65 | 0.6 |
| **Avg Success Rate** | 0.77 | **0.87** | 0.83 | 0.72 | 0.83 | 0.81 |

*Table 12.* Albation results on Training with $L_1$ (MAE) and $L_2$ (MSE) norm objective

| Task | Training Objective of Diffusion Bridge | |
|---|---|---|
| | $L_1$ norm (MAE) | $L_2$ norm (MSE) |
| Adroit Pen | 0.58 | 0.56 |
| Adroit Door | 0.84 | 0.83 |
| Adroit hammer | 1.00 | 1.00 |
| Avg | 0.81 | 0.80 |

*Table 13.* Success Rates of Individual Task on Adroit and DexArt benchmarks.

| Methods\Tasks | Adroit | | | DexArt | | | |
|---|---|---|---|---|---|---|---|
| | hammer | door | pen | laptop | faucet | toilet | bucket |
| DP | 0.45±0.04 | 0.35±0.03 | 0.15±0.03 | 0.69±0.06 | 0.23±0.06 | 0.63±0.06 | 0.23±0.06 |
| DP3 | **1.00±0.00** | 0.65±0.04 | 0.40±0.04 | 0.86±0.06 | 0.36±0.03 | 0.73±0.05 | **0.32±0.05** |
| Simple DP3 | **1.00±0.00** | 0.59±0.01 | 0.45±0.02 | 0.79±0.05 | 0.26±0.03 | 0.63±0.05 | 0.22±0.05 |
| FlowPolicy | **1.00±0.00** | 0.58±0.04 | 0.53±0.08 | 0.80±0.04 | 0.38±0.06 | 0.70±0.04 | 0.28±0.04 |
| VITA | 0.96±0.01 | 0.81±0.03 | 0.55±0.04 | 0.82±0.04 | 0.39±0.03 | 0.72±0.04 | 0.28±0.04 |
| BridgePolicy | **1.00±0.00** | **0.84±0.05** | **0.58±0.07** | **0.88±0.04** | **0.44±0.05** | **0.76±0.05** | **0.32±0.05** |

*Table 14.* Success Rates of Individual Task on MetaWorld (Easy) benchmark.

| Methods\Tasks | MetaWorld (Easy) | | | | | | |
|---|---|---|---|---|---|---|---|
| | button-press | button-press-topdown | button-press-topdown-wall | button-press-wall | coffee-button | dial-turn | door-close |
| DP | 0.93±0.01 | 0.98±0.01 | 0.96±0.02 | 0.93±0.02 | 0.99±0.01 | 0.63±0.07 | **1.00±0.00** |
| DP3 | **1.00±0.00** | **1.00±0.00** | **0.99±0.01** | 0.99±0.01 | **1.00±0.00** | 0.66±0.01 | **1.00±0.00** |
| Simple DP3 | **1.00±0.00** | **1.00±0.00** | 0.98±0.01 | 0.93±0.01 | 1.00±0.00 | 0.59±0.01 | 0.97±0.00 |
| FlowPolicy | **1.00±0.00** | 0.97±0.02 | 0.98±0.01 | **1.00±0.00** | **1.00±0.00** | 0.88±0.07 | 0.90±0.06 |
| VITA | 0.99±0.01 | **1.00±0.00** | 0.99±0.01 | **1.00±0.00** | **1.00±0.00** | **0.90±0.14** | **1.00±0.00** |
| BridgePolicy | **1.00±0.00** | **1.00±0.00** | 0.93±0.01 | **1.00±0.00** | **1.00±0.00** | 0.83±0.01 | **1.00±0.00** |

*Table 15.* Success Rates of Individual Task on MetaWorld (Easy) benchmark.

| Methods\Tasks | MetaWorld (Easy) | | | | | | |
|---|---|---|---|---|---|---|---|
| | door-lock | door-open | lever-pull | drawer-close | reach | window-close | window-open |
| DP | 0.75±0.06 | 0.98±0.02 | 0.98±0.02 | **1.00±0.00** | 0.93±0.02 | 0.99±0.01 | **1.00±0.00** |
| DP3 | 0.98±0.01 | 0.86±0.01 | **1.00±0.00** | **1.00±0.00** | **1.00±0.00** | **1.00±0.00** | **1.00±0.00** |
| Simple DP3 | 0.98±0.01 | 0.76±0.06 | **1.00±0.00** | **1.00±0.00** | **1.00±0.00** | **1.00±0.00** | **1.00±0.00** |
| FlowPolicy | **1.00±0.00** | 0.66±0.06 | **1.00±0.00** | 0.75±0.01 | **1.00±0.00** | 0.76±0.06 | 0.73±0.04 |
| VITA | 0.90±0.01 | 0.95±0.01 | 0.68±0.05 | **1.00±0.00** | 0.19±0.00 | 0.76±0.04 | 0.73±0.08 |
| BridgePolicy | **1.00±0.00** | **1.00±0.00** | 0.77±0.01 | **1.00±0.00** | 0.36±0.01 | **1.00±0.00** | **1.00±0.00** |

*Table 16.* Success Rates of Individual Task on MetaWorld (Easy) benchmark.

| Methods\Tasks | MetaWorld (Easy) | | | | | | |
|---|---|---|---|---|---|---|---|
| | hand-pull | hand-pull-side | door-unlock | peg-unplug-side | drawer-open | reach-wall | plate-slide-back |
| DP | 0.75±0.06 | **0.98±0.02** | 0.98±0.02 | **1.00±0.00** | 0.93±0.02 | 0.99±0.01 | **1.00±0.00** |
| DP3 | 0.98±0.01 | 0.86±0.01 | **1.00±0.00** | **1.00±0.00** | **1.00±0.00** | 1.00±0.00 | **1.00±0.00** |
| Simple DP3 | 0.98±0.01 | 0.76±0.06 | **1.00±0.00** | **1.00±0.00** | **1.00±0.00** | **1.00±0.00** | **1.00±0.00** |
| FlowPolicy | **1.00±0.00** | 0.66±0.06 | **1.00±0.00** | 0.75±0.01 | **1.00±0.00** | 0.76±0.06 | 0.73±0.04 |
| VITA | 0.34±0.07 | 0.54±0.07 | 0.99±0.01 | 0.81±0.08 | **1.00±0.00** | 0.71±0.05 | 0.99±0.01 |
| BridgePolicy | **1.00±0.00** | 0.92±0.01 | **1.00±0.00** | 0.93±0.03 | **1.00±0.00** | 0.80±0.04 | **1.00±0.00** |

*Table 17.* Success Rates of Individual Task on MetaWorld (Easy) and MetaWorld (Medium) benchmarks.

| Methods\Tasks | MetaWorld (Easy) | | | MetaWorld (Medium) | | | |
| --- | --- | --- | --- | --- | --- | --- | --- |
| | plate-slide-back-side | plate-slide | plate-slide-side | basketball | bin-picking | box-close | hammer |
| DP | 0.99±0.00 | 0.75±0.03 | **1.00±0.00** | 0.85±0.04 | 0.15±0.03 | 0.30±0.04 | 0.15±0.04 |
| DP3 | **1.00±0.00** | **1.00±0.01** | 1.00±0.00 | 0.98±0.01 | 0.38±0.21 | 0.42±0.02 | 0.72±0.03 |
| Simple DP3 | 0.99±0.00 | **1.00±0.01** | 1.00±0.00 | 0.95±0.03 | 0.28±0.10 | 0.38±0.04 | 0.62±0.06 |
| FlowPolicy | 0.65±0.04 | **1.00±0.00** | 1.00±0.00 | 0.66±0.04 | **0.66±0.10** | **0.81±0.03** | **0.98±0.01** |
| VITA | **1.00±0.00** | 0.96±0.02 | 1.00±0.00 | 0.90±0.01 | 0.23±0.01 | 0.52±0.04 | 0.90±0.01 |
| BridgePolicy | **1.00±0.00** | 1.00±0.00 | 1.00±0.00 | 1.00±0.00 | 0.45±0.04 | 0.80±0.06 | 0.95±0.05 |

*Table 18.* Success Rates of Individual Task on MetaWorld (Medium) benchmark.

| Methods\Tasks | MetaWorld (Medium) | | | | | | |
| --- | --- | --- | --- | --- | --- | --- | --- |
| | peg-insert-side | push-wall | soccer | coffee-pull | coffee-push | sweep | sweep-into |
| DP | 0.34±0.05 | 0.20±0.02 | 0.14±0.03 | 0.34±0.05 | 0.67±0.03 | 0.18±0.06 | 0.10±0.03 |
| DP3 | 0.67±0.05 | 0.51±0.06 | 0.18±0.02 | 0.85±0.02 | **0.94±0.02** | 0.96±0.02 | 0.17±0.04 |
| Simple DP3 | 0.48±0.05 | 0.38±0.06 | 0.16±0.02 | 0.92±0.08 | 0.86±0.04 | 0.88±0.02 | 0.09±0.04 |
| FlowPolicy | 0.70±0.06 | 0.63±0.06 | **0.33±0.04** | 0.96±0.01 | 0.61±0.04 | 0.70±0.03 | 0.31±0.01 |
| VITA | 0.42±0.08 | 0.51±0.10 | 0.19±0.03 | **1.00±0.00** | 0.63±0.04 | 0.87±0.03 | 0.12±0.01 |
| BridgePolicy | **0.77±0.04** | **0.87±0.02** | 0.31±0.04 | 0.97±0.01 | 0.87±0.02 | **0.97±0.01** | **0.40±0.02** |

*Table 19.* Success Rates of Individual Task on MetaWorld (Hard) benchmark.

| Methods\Tasks | MetaWorld (Hard) | | | | |
| --- | --- | --- | --- | --- | --- |
| | pick-out-of-hole | pick-place | assembly | push | hand-insert |
| DP | 0.00±0.00 | 0.02±0.01 | 0.15±0.01 | 0.28±0.02 | 0.09±0.01 |
| DP3 | 0.14±0.06 | 0.12±0.03 | 0.99±0.01 | 0.62±0.02 | 0.14±0.03 |
| Simple DP3 | 0.08±0.04 | 0.12±0.04 | 0.79±0.01 | 0.32±0.02 | 0.12±0.04 |
| FlowPolicy | **0.31±0.03** | 0.63±0.04 | **1.00±0.00** | 0.73±0.04 | **0.26±0.01** |
| VITA | 0.23±0.04 | 0.43±0.07 | 0.99±0.01 | 0.65±0.03 | 0.09±0.01 |
| BridgePolicy | 0.28±0.04 | **0.65±0.01** | **1.00±0.00** | **0.74±0.02** | 0.25±0.04 |

*Table 20.* Success Rates of Individual Task on MetaWorld (Very Hard) benchmark.

| Methods\Tasks | MetaWorld (Very Hard) | | | | |
| --- | --- | --- | --- | --- | --- |
| | stick-push | stick-pull | shelf-place | pick-place-wall | disassemble |
| DP | 0.63±0.02 | 0.11±0.01 | 0.11±0.02 | 0.05±0.01 | 0.43±0.05 |
| DP3 | 0.97±0.03 | 0.27±0.06 | 0.19±0.07 | 0.35±0.06 | 0.75±0.03 |
| Simple DP3 | 0.97±0.04 | 0.15±0.06 | 0.05±0.01 | 0.28±0.04 | 0.50±0.02 |
| FlowPolicy | **1.00±0.00** | 0.56±0.02 | **0.40±0.02** | 0.95±0.02 | 0.88±0.02 |
| VITA | 0.83±0.05 | 0.61±0.02 | 0.16±0.04 | 0.80±0.06 | 0.69±0.02 |
| BridgePolicy | **1.00±0.00** | **0.82±0.02** | 0.25±0.02 | **0.96±0.04** | **0.92±0.02** |

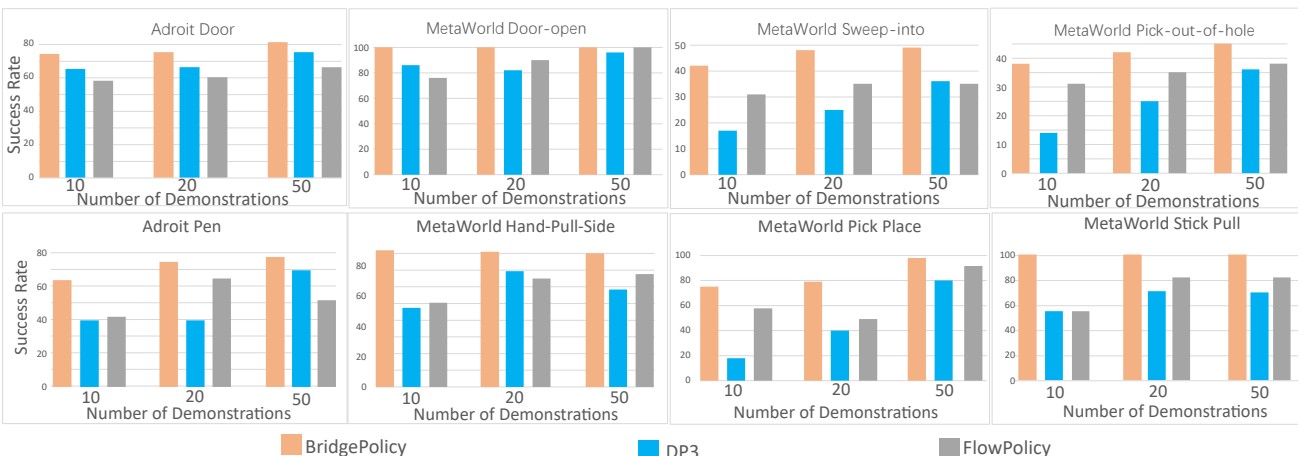

*Figure 15.* Additional sample efficiency ablation results.

