# OpenReview forum: "Sample from What You See: Visuomotor Policy Learning via Diffusion Bridge with Observation-Embedded Stochastic Differential Equation"
_ICML.cc/2026/Conference — ICML 2026 regular_

### Official Review · Reviewer_Sn4r · 2026-03-10

**Soundness:** 2
**Presentation:** 2
**Significance:** 2
**Originality:** 1
**Overall Recommendation:** 4
**Confidence:** 4

**Summary:**

This paper focuses on applying diffusion bridge models to visuomotor policy learning for robotic imitation learning, proposing the BridgePolicy framework that embeds observations into diffusion SDE trajectories to sample actions from an observation-informed prior instead of random noise with a multi-modal fusion module and a semantic aligner to resolve modality and shape mismatches between robotic observations and actions.

**Compliance With Llm Reviewing Policy:**

Affirmed.

**Final Justification:**

The reply does fully address my core concerns. Therefore, I would raise my score as weakly accept.

**Key Questions For Authors:**

1. The core diffusion bridge framework is fully reused from UniDB (Zhu et al., 2025) without theoretical modifications. Please specify the concrete theoretical innovations compared to UniDB.
2. Multiple success rate results in Table 1 have too large standard deviation. Please correct these statistical errors and provide accurate, bounded success rate results for all tasks.
3. The experiment uses an unfair NFE setting (1 for FlowPolicy, 10 for the proposed method). Please supplement head-to-head comparison results of all baselines under the same 10-step NFE in the main text and explain the original inconsistent setting.

**Limitations:**

Yes

**Strengths And Weaknesses:**

**Strengths:**
1. The paper includes mainstream baselines, basic ablation studies and complete implementation details in the appendix, with a mathematically self-consistent proof for  theoretical guarantee.
2. The paper empirically verifies the feasibility of applying diffusion bridge models to visuomotor policy learning, providing experimental references for the relevant subfield.
3. The paper is the first work to adapt the UniDB diffusion bridge framework to visuomotor policy learning tasks.

**Weaknesses:**
1. The paper lacks sufficient innovation. The core diffusion bridge framework is fully reused from prior work without theoretical innovation with insufficient design justification for core modules and unclear definitions of some key mathematical symbols.
2. The paper has flaws in its experimental design and result. Core success rate results have excessively large standard deviations; baseline comparisons are unfair due to inconsistent NFE settings.
3. The so-called novel fusion and alignment modules are built on mature off-the-shelf techniques, and all claimed innovations are just routine engineering implementations .

---

> ### Author Rebuttal · Authors · 2026-03-31
>
> We sincerely thank you for your constructive comments and questions.
>
> > W1: The paper lacks sufficient innovation. The core diffusion bridge framework is fully reused from prior work without theoretical innovation with insufficient design justification for core modules and unclear definitions of some key mathematical symbols.
>
> > Q1: The core diffusion bridge framework is fully reused from UniDB (Zhu et al., 2025) without theoretical modifications. Please specify the concrete theoretical innovations compared to UniDB.
>
> > W3: The so-called novel fusion and alignment modules are built on mature off-the-shelf techniques, and all claimed innovations are just routine engineering implementations.
>
> A: We first clarify our claim precisely: BridgePolicy does not propose a new general diffusion-bridge theory beyond UniDB. **Our contribution is instead a new policy-learning formulation and analysis that make diffusion bridge usable for visuomotor control**, which extends the diffusion bridge formulation to broader applications. Specifically:
>
> * **Observation-embedded in Diffusion SDE formulation**. Unlike previous diffusion policies that treat observations only as external conditions in the neural network and sample from uninformative Gaussian noise, we reformulate generative policy learning via diffusion bridge so that the reverse process starts from an observation-informed endpoint $𝑎_𝑇=𝑧_{obs}$. This changes the role of observations from conditional inputs to variables explicitly embedded in the SDE trajectory, which is the key modeling difference of BridgePolicy.
>
> * **Making diffusion bridge workable for heterogeneous robot observation**. Existing diffusion bridge models must require the two endpoints to have the same data shapes. In robotics, however, observations and actions are inherently multi-modal, semantically heterogeneous, and shape-mismatched, making the diffusion bridge **cannot be directly applied off the shelf**. Our designed fusion module and aligner are introduced exactly to construct a bridgeable observation representation for policy learning. This is not a superficial architectural addition; it is the mechanism that makes diffusion bridge feasible in the robotic setting.
>
> * **Task-specific theoretical analysis beyond direct reuse.** Theorem 3.1 provides a robustness result showing that perturbations in the learned observation representation induce only bounded deviations in the generated action. This analysis is specific to our observation-to-action bridge formulation.
>
> > W2: The paper has flaws in its experimental design and result. Core success rate results have excessively large standard deviations; baseline comparisons are unfair due to inconsistent NFE settings.
>
> > Q2: Multiple success rate results in Table 1 have too large standard deviation. Please correct these statistical errors and provide accurate, bounded success rate results for all tasks.
>
> A: Following FlowPolicy, we group tasks by benchmark and difficulty level, and report results accordingly. The standard deviations in Table 1 are the **deviations aggregated over all tasks** in the same benchmark or difficulty group, **not the variation of one task across different runs**. Since tasks within the same group can vary in **difficulty**, the aggregated deviations can be relatively large. The actual mean ± standard deviation for each individual task over 3 random seeds are **already reported in Appendix Tables 11–18**. We will change the statistics for deviation to statistics based on 3 random seeds.
>
> > Q3: The experiment uses an unfair NFE setting (1 for FlowPolicy, 10 for the proposed method). Please supplement head-to-head comparison results of all baselines under the same 10-step NFE in the main text and explain the original inconsistent setting.
>
> A: All baselines except FlowPolicy are already **evaluated with NFE=10** in the main text. For FlowPolicy, we followed the original paper and used NFE=1, because multi-step sampling in consistency flow matching suffers from error accumulation, which may harm the performance [R1]. Importantly, we **have already increased the NFE to 10 and provided the results in Appendix E.2/Table 6**: the success rate reduces from 0.64 to 0.59. Therefore, the conclusion is unchanged under the same 10-step setting. We will move this clarification and the corresponding results to the main text to avoid confusion.
>
> [R1] Sabour et al. "Align your flow: Scaling continuous-time flow map distillation", NeurIPS 2025.

---

> > ### Author Rebuttal · Reviewer_Sn4r · 2026-04-03
> >
> > > (Q2) Statistical Deviations
> >
> > I appreciate the authors pointing me to Appendix Tables 11-18 for the per-task statistics. However, upon carefully reviewing these tables, I found a clear mathematical contradiction in the reported statistics that raises concerns about data integrity.In several instances, the success rate is reported as having a mean of 1.00 but a non-zero standard deviation. For example, in Table 13, BridgePolicy on the 'door-lock' task is reported as $1.00 \pm 0.02$. Could the authors please explain this mathematical impossibility?
> >
> > > (Q3) NFE Settings for Baselines
> >
> > The justification for evaluating FlowPolicy at NFE=1 is technically sound and well-supported by the cited literature on consistency flow matching. Furthermore, pointing out the 10-step degradation in Appendix Table 6 effectively resolves my concern about unfair comparisons.
> >
> > > (Q1 & W1 & W3) Clarification on Contributions
> >
> > I agree that while the fundamental mathematics strictly follow UniDB, reformulating the SDE endpoints to $a_T = z_{obs}$ and bridging the heterogeneous modality gap between visual observations and proprioceptive actions are non-trivial challenges in visuomotor control. Consequently, I maintain my original assessment that the strictly theoretical innovation is limited.
> >
> > ---
> > Overall, the experimental evaluation and the ablations presented in this paper are quite comprehensive. If the authors can address my concerns regarding the data integrity and provide a clear explanation for these mathematical contradictions, I will certainly consider raising my Overall Recommendation score.

---

> > > ### Author Response · Authors · 2026-04-03
> > >
> > > Thanks for recognizing our clarifications. Here we reply to your remaining concern about the data integrity.
> > >
> > >  The $1.00 \pm 0.02$ is a typo. We check the evaluation log, and it should be $1.00 \pm 0.00$. We will correct it in the revision. For other cases, the success rate $1.00 \pm 0.01$ of DP3 and Simple DP3 for plate-slide tasks is due to the rounding approximation. For instance, if we use three random seeds and the corresponding success rates are $ 1.00, 1.00, 0.99$, then this yields $1.00 \pm 0.01$. This phenomenon was also observed in Table XVIII of DP3 [R1].
> > >
> > > [R1] Ze et al. "3D Diffusion Policy: Generalizable Visuomotor Policy Learning via Simple 3D Representations", RSS 2024.

---

### Official Review · Reviewer_WuYy · 2026-03-10

**Soundness:** 3
**Presentation:** 3
**Significance:** 3
**Originality:** 3
**Overall Recommendation:** 4
**Confidence:** 4

**Summary:**

This paper proposes BridgePolicy, a generative framework for visuomotor policy learning. Unlike prior diffusion-based policies that mainly use observations as conditional inputs to the denoising network, the proposed method explicitly embeds observations into the stochastic differential equation trajectory through a diffusion-bridge formulation, so inference starts from an informative observation-induced prior rather than uninformative Gaussian noise. To address heterogeneity and shape mismatch between robotic observations and actions, the paper further introduces a multimodal fusion module and a semantic aligner that fuse point clouds and robot states into an action-compatible representation space. Experiments on large-scale simulation tasks and 5 real-world robot tasks show that the method outperforms several existing generative policy baselines overall, with particularly strong performance on tasks requiring precise manipulation.

**Compliance With Llm Reviewing Policy:**

Affirmed.

**Final Justification:**

The paper proposes a meaningful adaptation of diffusion bridge modeling to visuomotor policy learning. Its main strength lies in explicitly incorporating observations into the generation trajectory, rather than using them only as external conditions, and in introducing practical modules for multimodal fusion and alignment. The method is technically sound and supported by fairly broad experiments in both simulation and real-robot settings.

My main concerns remain the limited theoretical novelty beyond prior bridge-based work, as well as the relatively limited statistical evidence in the experiments. The rebuttal was helpful in clarifying the paper’s intended contribution, fixing the task-count inconsistency, and adding one robustness result under simulated occlusion. However, these responses only partially addressed my concerns and did not materially change my overall assessment.

Overall, I still view this as a solid and relevant contribution with clear practical value, despite some limitations in theoretical depth and experimental coverage. For this reason, I maintain my original weak accept recommendation.

**Key Questions For Authors:**

1. The empirical study around Theorem 3.1 suggests a small constant $C$, but would this still hold under more realistic perturbations such as point-cloud occlusion, calibration error, or depth noise? I would appreciate robustness experiments that better reflect sensor imperfections.

2. The paper currently relies on multi-step sampling from an observation prior; how do the authors view future integration with consistency distillation or other one-step policy generation methods? Which parts of the current bridge formulation are the main obstacles?

3. The method appears strong on short-horizon action chunks, but how well does it extend to longer-horizon, multi-stage tasks? When task execution requires stage transitions or longer planning horizons, could anchoring the bridge endpoint to the current observation limit action continuity or foresight?

**Limitations:**

Yes.

**Strengths And Weaknesses:**

**Strengths:**

1. The paper formulates policy learning as a diffusion bridge rather than a standard conditional diffusion model, and the idea of sampling actions from an observation-informed prior is well motivated both mathematically and conceptually.

2. Beyond the bridge formulation itself, the paper introduces multimodal fusion and semantic alignment modules to address the mismatch between heterogeneous observations and action representations in robotics, making the approach practically applicable.

3. The paper evaluates the method on multiple simulation benchmarks and several real-world tasks, and includes ablations on modality fusion, aligner weight, sample efficiency, and comparisons against regression and standard diffusion heads.

4. On real-robot tasks, the method achieves stronger average success rates than DP3 and FlowPolicy, with intuitive qualitative improvements on tasks such as oven opening and unplugging that require precise control.

**Weaknesses:**

1. The theoretical foundation is largely adapted from prior bridge work rather than being entirely new bridge theory. The mathematical bridge framework and fast sampling procedure rely substantially on UniDB / UniDB++, while the main contribution here is the adaptation to visuomotor policy learning and the additional components for heterogeneous robot data.

2. Theorem 3.1 does not by itself establish strong robustness to real sensor noise or overall system stability. It relies on a Lipschitz assumption on the prediction network and gives a linear upper bound from terminal perturbation to action error.

3. The paper reports the mean of the top-5 success rates per seed, so the reported numbers may not fully reflect run-to-run variability. In addition, the real-world evaluation uses only 10 trials per task, so the statistical evidence is still limited.

4. Although a fast sampler is used, BridgePolicy still relies on multi-step iterative inference. The paper also notes that the current SDE formulation limits compatibility with one-step distillation.

5. The manuscript is inconsistent about the total number of simulation tasks ("52 tasks" in the text versus "50 tasks" in a table caption), which makes the experimental setup a bit harder to follow.

---

> ### Author Rebuttal · Authors · 2026-03-31
>
> Thank you for your constructive feedback.
>
> > W1: The theoretical foundation is largely adapted from prior bridge work rather than being entirely new bridge theory. The mathematical bridge framework and fast sampling procedure rely substantially on UniDB / UniDB++, while the main contribution here is the adaptation to visuomotor policy learning and the additional components for heterogeneous robot data.
>
> A: The diffusion bridge theory and fast sampling algorithm are built on previous theoretical work UniDB and UniDB++. Our novelty is on the policy modeling side: we **reformulate generative policy learning so that observations are explicitly embedded into the diffusion SDE trajectory**, rather than used only as external conditions. This enables the generation to sample actions from an observation-informed prior instead of random noise. Additionally, we bridge the shape and modality gap hindering the application of diffusion bridge to policy learning by introducing a fusion-and-aligner module.
>
> > W2: Theorem 3.1 does not by itself establish strong robustness to real sensor noise or overall system stability. It relies on a Lipschitz assumption on the prediction network and gives a linear upper bound from terminal perturbation to action error.
>
> > Q1: The empirical study around Theorem 3.1 suggests a small constant, but would this still hold under more realistic perturbations such as point-cloud occlusion, calibration error, or depth noise? I would appreciate robustness experiments that better reflect sensor imperfections.
>
> A: Theorem 3.1 discusses the **impact of the fitting error** introduced by the aligner **on the generated actions** of diffusion bridge. The depended **Lipschitz assumption is widely adopted** in prior diffusion-solver analyses [R1, R2] and proved to be solid.
>
> For the concern of sensor imperfections, **we conduct an experiment on the MetaWorld plate-slide that simulates the possible real sensor error**. Specifically, we first randomly select a cluster of 50 points in the point cloud (out of 512 points) and replace it with a plane to simulate point cloud occlusion. The success rate drops from 1.0 to 0.85, while FlowPolicy drops from 1.0 to 0.75, which indicates BridgePolicy could maintain a relatively higher success rate under occlusion, demonstrating its robustness to point cloud imperfections.
>
> > W4: Although a fast sampler is used, BridgePolicy still relies on multi-step iterative inference. The paper also notes that the current SDE formulation limits compatibility with one-step distillation.
>
> > Q2: The paper currently relies on multi-step sampling from an observation prior; how do the authors view future integration with consistency distillation or other one-step policy generation methods? Which parts of the current bridge formulation are the main obstacles?
>
> A: As discussed in the Conclusion, the current BridgePolicy still relies on iterative sampling. The main obstacle is that the SDE formulation prevents many distillation methods from being used, thus forbidding the one-step sampling. To enable the one-step sampling, an ODE formulation of BridgePolicy must first be derived. We're willing to explore it as our future work.
>
> > W3: The paper reports the mean of the top-5 success rates per seed, so the reported numbers may not fully reflect run-to-run variability. In addition, the real-world evaluation uses only 10 trials per task, so the statistical evidence is still limited.
>
> A: The evaluation of BridgePolicy directly follows the standard protocol used in prior generative policy work for fair comparison (like DP3 and FlowPolicy). We would be happy to include additional metrics to reflect run-to-run variability in revision.
>
> > W5: The manuscript is inconsistent about the total number of simulation tasks ("52 tasks" in the text versus "50 tasks" in a table caption), which makes the experimental setup a bit harder to follow.
>
> A: It is a typo that the total simulation task number should be 52. Thanks for pointing it out and we will fix it in the revision.
>
> > Q3: The method appears strong on short-horizon action chunks, but how well does it extend to longer-horizon, multi-stage tasks? When task execution requires stage transitions or longer planning horizons, could anchoring the bridge endpoint to the current observation limit action continuity or foresight?
>
> A: Long-horizon and multi-stage execution generally calls for additional architectural components, such as codebook-based mechanisms, to capture higher-level temporal structure. Since these components are largely modular, they can be integrated into BridgePolicy just as they can be introduced into Diffusion Policy. We would like to leave this promising direction as our future work.
>
> [R1] Lu et al. "Dpm-solver: A fast ode solver for diffusion probabilistic model sampling in around 10 steps", NeurIPS 2022.
>
> [R2] Lu et al. "Dpm-solver++: Fast solver for guided sampling of diffusion probabilistic models, Machine Intelligence Research 2025.

---

> > ### Author Rebuttal · Reviewer_WuYy · 2026-04-01
> >
> > The rebuttal clarifies several points and resolves the task-count typo. It also adds one useful robustness result. However, these updates do not substantially change my overall assessment, so I will keep my current score.

---

> > > ### Author Response · Authors · 2026-04-01
> > >
> > > Thank you for acknowledging our clarification, recognizing the value of the additional robustness results, and for **maintaining a positive rating.**

---

### Official Review · Reviewer_kvsp · 2026-03-13

**Soundness:** 3
**Presentation:** 3
**Significance:** 3
**Originality:** 2
**Overall Recommendation:** 4
**Confidence:** 3

**Summary:**

This paper proposes BridgePolicy, a visuomotor policy that transfers the idea of Diffusion Bridges to robotic manipulation, enabling the policy to directly bridge from the distribution of state observations to the target action distribution. The authors introduce a multimodal fusion module and a semantic aligner to address challenges such as heterogeneous observations and the lack of natural alignment between observations and actions. The method is evaluated on 52 tasks, and the results demonstrate its effectiveness.

**Compliance With Llm Reviewing Policy:**

Affirmed.

**Final Justification:**

I appreciate the detailed response. The clarifications and new results provided have resolved the majority of my concerns.

**Key Questions For Authors:**

What is the potential for extending BridgePolicy to VLA settings? This relates to two aspects. First, the input feature representation differs. Second, how does BridgePolicy perform with a small number of NEFs, given that many flow-matching approaches in VLA only require a few NEFs (e.g., 4)?

**Limitations:**

Yes

**Strengths And Weaknesses:**

- Strengths
  1. The paper is well written—clear and easy to follow—and provides complete theoretical formulations.
  2. This is the first work to adapt Diffusion Bridges as a visuomotor policy for robotic tasks, offering an additional research perspective for the field.
  3. The authors design a set of mechanisms for modality integration and alignment, which enables a more effective application of the Diffusion Bridge framework.
  4. Overall, BridgePolicy achieves better performance than several classical visuomotor policies.

- Weaknesses
  1. Clarification of differences from prior work. I would like the authors to further elaborate on how this approach differs from VITA, which also directly maps vision to actions via a flow-matching framework. In particular, why does formulating the Diffusion Bridge as an SDE yield a significant advantage over the ODE formulation adopted in VITA?
  2. Code release. Given the claimed effectiveness of the proposed Diffusion Bridge approach, I encourage the authors to release their code to increase the contribution and reproducibility for the community.
  3. The ablation study does not seem to examine the impact of Modality Alignment. Does this component contribute substantially to BridgePolicy’s performance?
  4. It would also be helpful to include an experiment where the observation fusion pipeline is made identical to prior methods (e.g., DP, DP3, and Flow Matching) and Modality Alignment is removed. Such a controlled comparison would better isolate and demonstrate the advantage of the Diffusion Bridge formulation itself.

---

> ### Author Rebuttal · Authors · 2026-03-31
>
> Thank you for your constructive feedbacks.
>
> > W1: Clarification of differences from prior work. I would like the authors to further elaborate on how this approach differs from VITA, which also directly maps vision to actions via a flow-matching framework. In particular, why does formulating the Diffusion Bridge as an SDE yield a significant advantage over the ODE formulation adopted in VITA?
>
> A: Our method differs from VITA in two fundamental ways.
>
> * First, **VITA** adopted a **two-stage** training way while **BridgePolicy** was trained **end-to-end**, which generate executable actions directly. VITA perform flow matching in a joint image-action latent space and its performance largely depends on the quality of the latent space. It requires an additional decoder to recover executable actions, which could introduce additional errors.
> * Second, BridgePolicy is built upon SDE-based diffusion bridge while VITA is upon ODE-based flow matching. Recent work [R1] shows that **Diffusion Bridge** constructs a **more stable transition** between two arbitrary distributions and demonstrates better performance on large distributional shifts (e.g. heterogeneous observations and actions) **than Flow Matching**. Our experiments also show our superior success rates in both simulation and real-world tasks.
>
> > W2: Code release. Given the claimed effectiveness of the proposed Diffusion Bridge approach, I encourage the authors to release their code to increase the contribution and reproducibility for the community.
>
> A: Thank you for the suggestion. In accordance with the ICML Anonymity and Links policy, we have shared the anonymous code repository with the AC instead of attaching it here. The code can be accessed through the AC for review purpose. We commit to releasing the code upon acceptance.
>
> > W3: The ablation study does not seem to examine the impact of Modality Alignment. Does this component contribute substantially to BridgePolicy’s performance?
>
> A: Modality Alignment is a necessary component of BridgePolicy, otherwise the BridgePolicy does not work. Diffusion bridge requires **endpoints must be in the same dimension** (aligner reshape the observaiton representation to enable the diffusion bridge SDE calculation), thus a complete removal of the aligner is not suitable for diffusion bridge formulation.
>
> > W4: It would also be helpful to include an experiment where the observation fusion pipeline is made identical to prior methods (e.g., DP, DP3, and Flow Matching) and Modality Alignment is removed. Such a controlled comparison would better isolate and demonstrate the advantage of the Diffusion Bridge formulation itself.
>
> A: As stated in **the Answer to W3**, Modality Alignment cannot be removed for BridgePolicy. We demonstrated **the advantages of diffusion bridge formulation in Table 5**, where we keep the modality fusion and aligner of BridgePolicy unchanged and only replace its policy formulation from Diffusion Bridge to Diffusion. Under this setting, Diffusion Bridge consistently performs best, e.g., Adroit Door 0.84 vs. 0.68 for standard Diffusion, and MW Coffee Pull 0.97 vs. 0.86.
>
> > Q1: What is the potential for extending BridgePolicy to VLA settings? This relates to two aspects. First, the input feature representation differs. Second, how does BridgePolicy perform with a small number of NEFs, given that many flow-matching approaches in VLA only require a few NEFs (e.g., 4)?
>
> A: We view extension to VLA as a promising future direction, it is not yet validated in this paper. The same observation encoding principle of BridgePolicy could be applied to vision-language features by projecting them into the action-aligned representation. Regarding small NFEs, BridgePolicy could reduce NFE to 5. We conduct **additional experiments** and find that reducing NFEs from 10 to 5 only slightly drop of the average success rate on Adroit (0.81 to 0.79), which suggests encouraging potential for low-NFE settings as well. We will clarify this discussion in the revision.
>
> [R1] Zhu et al. "Diffusion Bridge or Flow Matching? A Unifying Framework and Comparative Analysis", arXiv 2025.

---

> > ### Author Rebuttal · Reviewer_kvsp · 2026-04-03
> >
> > I appreciate the detailed response. The clarifications and new results provided have resolved the majority of my concerns. I have updated my score to reflect this.

---

> > > ### Author Response · Authors · 2026-04-03
> > >
> > > Thank you for acknowledging our clarification and raising the score!

---

### Official Review · Reviewer_z5Nz · 2026-03-13

**Soundness:** 4
**Presentation:** 3
**Significance:** 3
**Originality:** 2
**Overall Recommendation:** 5
**Confidence:** 4

**Summary:**

BridgePolicy is a robotic imitation learning policy, compared to standard diffusion/flow policies which start from noise, BridgePolicy uses a diffusion bridge connecting the observation distribution to the action one, so that the generative process starts from the observation instead of a Gaussian noise. For the shapes to match the authors first process the observations (point cloud and state) to a latent representation, that is aligned to the action distribution by a contrastive learning loss.  Through experiments and ablation studies, on both simulation and real-world tasks, the authors demonstrate the effectiveness of their approach.

**Compliance With Llm Reviewing Policy:**

Affirmed.

**Final Justification:**

My original grade was *Weak accept*. After some hesitation, I ultimately decided to upgrade it to *Accept*.

While I share the other reviewers' concerns about the manuscript's lack of clarity regarding the novelty and its strong phrasing. I trust the authors to correct their phrasing and clarify that the novelty is not the diffusion bridge framework itself, but its application to robotics control. Their efforts during the rebuttal confirmed my confidence in the soundness, especially their new ablation study (despite some crucial missing details in their first response...).

**Key Questions For Authors:**

- What are the hyperparameters for simulation tasks, as Table 7 is only for real world tasks ? Is alpha tuned per-task ? (see W3.1)
- As explained W2 "multi-modal" has two meanings. Is BridgePolicy able to generate multi-modal action distributions, despite the one-to-one mapping between observations and actions ? (as a small comment: an experiment for instance on the Push-T task as in Diffusion Policy (Chi et al) would have clarified it)
- What is the motivation of using an l1 norm Eq 3 ? And where is the ablation study mentioned ?
- From the ablation study Fig 4, BridgePolicy seems especially good in low data regimes, achieving good scores with only 10 demonstrations. In the real-world setup the number of demonstrations is fixed to 50, did the authors try with less demonstrations ?
- Could the fundamental difference with Vita/Bridger (see W1) be more detailed?

If some of the questions are correctly addressed, I'm very inclined to increase my score to "Accept".

**Limitations:**

Yes

**Strengths And Weaknesses:**

## Strengths
- The paper is well written, the narrative is easy to follow. The Appendix is excellent, as it is clear, factual while sticking to what is relevant (only one part is missing, see W3.1).
- Proof of Theorem 3.1 in Appendix A is clear, and the associated experiment is relevant, except one hypothesis is missing (see W3.3).
- Overall the experiments and ablation studies are well done. Beyond the good ablation study on BridgePolicy (Tab 3, 4, 5 and Fig 4 and 12), Tab 6 is a good addition to reassure readers.
- The authors did real world experiments, and Appendix C is clean.

## Weaknesses
### W1
I find the narrative too strong on the fundamental differences in the way observations are treated compared to previous work. Saying that previously people were "*merely* treating (observations) as conditions" (line 082) and that in the proposed approach "the reserve process can *naturally* start from the observation", is too strong phrasing given that:
1) BridgePolicy's reverse process does not start directly from the observations, but from a preprocess of it; and it does add noise in the process (line 220).
2) Previous approaches often use inpainting in addition to the conditioning, so $a_T$ actually contains $o_s$ the observed state and so the reverse process partially starts from the observation.
3) BridgePolicy complete pipeline consists of first processing the observations through MLPs and then transforming it through a stochastic process that somehow interpolate this representation of the observations to the action, + some noise. So it is not too different from *Bridger* and *VITA*.

On this, some clarifications are needed about the exact training objective. In the backward process, is there a *stop gradient* to avoid modifying $\phi_1$ and $\phi_2$ (the parameters of the observation processing part) during training of the diffusion bridge part. Namely, is Eq 3 actually
$$\mathcal L_{DB} = \mathbb E_{t,o,a} [\| a_\theta(a_t^{\phi_1,\phi_2},a_T^{\phi_1,\phi_2},t) -a \|]$$
And is there a *stop-gradient* to avoid exploiting $\frac{\partial \mathcal L_{DB}}{\partial \phi_1}$ ? In case $\frac{\partial \mathcal L_{DB}}{\partial \phi_1}$ is exploited, it implies that the noised input can be modified by the reverse process and so MLP $\phi_1$ and $\phi_2$ can be considered as just processing the observations to condition the generative process.

Details are especially needed as in Appendix E.1, line 818 reports that the aligner's parameters ($\phi_1$ and $\phi_2$ ?) are frozen after 50 epochs, out of a total of 3000. This seems to correlate with the impression that modifying the starting point of the diffusion bridge might create instability, and so this fact should be mentioned and discussed in the paper.

### W2
"Multi-modal" has two different meanings and both are used consecutively, creating some confusion:
1) Multi-modal observation in the sens of different of different ways of getting some observation, using different medium (line 090, line 107 etc).
2) Multi-modal in the distributional sens, as mentioned in the first line of the abstract "capturing multi-modal action distributions".
These two meanings should be clarified and ideally it would be better to use different terms, in particular for the (1), "heterogeneous" seems enough ?

Currently this makes the paragraph line 089 to line 096 (right column) unclear. As it could be understood that there should be a mapping one-to-one between 1 obs and 1 action, in which case it would imply that unlike diffusion policies, BridgePolicy cannot generate multi-modal action distributions (with different actions possible from the same observation).

### Other comments / W3
- (W3.1) The hyperparameter table for the simulation tasks is missing. Currently only the hyperparameters for the real-world experiments are reported Table 7 (except if I missed it). In particular, it is not clear to me if the $\alpha$ parameter has been tuned per-task, or fixed to 1 for all tasks.
- (W3.2) Some paragraphs are repetitive about the general idea, for instance the end of Sec 2.1 just repeat the motivation and the plan again. "applying the diffusion bridge does not work the modality and shape mismatch...."
- (W3.3) Theorem 3.1, the fact that the sampling noise is the same for both predictions should be mentioned in the hypothesis, otherwise the theorem implies BridgePolicy cannot generate multi-modal action distributions. The last term of Eq 11 corresponds to this: the same noise $\epsilon$ is used in both prediction, but $\epsilon$ is not even defined in the proof.
- (W3.4) The "fusion" component is a bit oversold, in fact the processing of the state and point cloud is: downsample the point cloud, apply an MLP to each, cross attention. This is a simple direct processing, while the phrasing in the introduction/motivation/method may make it sound like a complicated novel mechanism.
- (W3.5) The purpose of repeating UniDB++ math equations in the core of the paper is unclear to me. Eq 4 for instance is exactly UniDB++ original paper (Mokai Pan et al)'s Algorithm 2. As the authors wrote "the updating rule seems very complex", so right now it may give the impression that the math part page 4 was written partially with the purpose of looking impressive. The contribution of BridgePolicy is not to derive these equations but to plug them in place of the standard DDPM schedule. I understand that given that UniDB is less common than DDPM, detailing the maths insures completeness, but the exact formulas might better fit in the Appendix. Similarly Fig 2 could be more lightweight without the exact big formula, but I understand it is a personal opinion.
- (W3.6) Some Tables should be reordered in Sec 4 and 5. For instance, Tab 3 is not referenced anywhere, because the wrong table is referenced line 434, hence Tab 3 is placed by LaTeX too early in the text. Also, I would recommend the authors to put Fig 3 next to Tab 2 instead of Tab 1 (to put together everything about real-world experiments).
- (W3.7) Reporting the highest score across the training epochs is a debated metric; it is now usually recommended to report the average score across the last few epochs. But this issue is attenuated as the authors take the five highest scores.

In summary, the paper is well done, and the experiments are solid, but the novelty could be summarized as:
1) using two MLPs + CrossAttn to process the observations
2) Then plug it in UniDB++, instead of the standard DDPM
3) Add an alignment loss to improve the processing part, but deactivate it after 50 epochs, out of 3000.
As highlighted in ICML guidelines, incorporating methods and applying them to a new field can be considered novelty and a good contribution. But I believe the narrative should be less strong on the complexity of the method / novelty of the method, and more as the application of UniDB++ to robotics, nearly as a plugging replacement for DDPM. Especially given that the experiments and ablation studies are good enough for the paper to be accepted.

Hence my current grade originality=2, because of ICML guideline "Are the contributions clearly distinguished from closely related literature" (also related to W1).

---

> ### Author Rebuttal · Authors · 2026-03-31
>
> We sincerely thank you for the constructive and detailed feedback.
> > W1: Strong phrasing.
>
> A: The reverse process indeed starts from the learned observation representation. We will remove the word "naturally" and use the "learned observation representation" to be more precise.
> > W1.3 & Q4: Fundamental difference with Vita/Bridger.
>
> A: We describe our differences from BRIDGER and VITA as follows:
> **BRIDGER** first obtain a **coarse policy** and then adopt diffusion bridge to **refine** the actions. Its performance largely depends on the quality of the coarse policy. In contrast, BridgePolicy does not depend on any prior policy.
> **VITA** adopts a **two-stage** pipeline. It performs flow matching in a **joint latent image-action space**, and the executable actions must be **recovered via a decoder**. BridgePolicy is trained **end-to-end** and directly generates the executable actions. On the other hand, BridgePolicy is built upon SDE-based diffusion bridge while VITA is upon ODE-based flow matching. Recent work [R1] shows that **diffusion bridge** constructs a **more stable transition** between two distributions and demonstrates better performance on large distributional shifts (e.g., observations and actions) **than flow matching**. Our experiments also show the superiority of our BridgePolicy compared to FlowPolicy in both simulation and real-world tasks. We will further discuss these differences in revision.
> [R1] Zhu et al. "Diffusion Bridge or Flow Matching? A Unifying Framework and Comparative Analysis", arXiv 2025.
> > Clarification on stop-gradient and additional training details.
>
> A: Thanks for such detailed comments. We didn't use stop-gradient during the training.
>
> The $\phi_1$ and $\phi_2$ serve as process observations to the condition in the generative process. At this time, the condition is still not aligned with the action in shape, thus cannot participate in the computation of the reverse SDE.
>
> The aligner here is another module other than $\phi_1$ and $\phi_2$ trained with the CLIP loss to reshape the condition into the actions' shape to enable the computation of the reverse SDE. We do not observe instability during the training even the aligner is trained for 3000 epochs. However, training the aligner throughout the entire process does slightly affect performance, so we use this trick. We will supplement more training details in the revision.
> > W2 & Q2: Usage of "multi-modal" and motivation of $l_1$ norm training objective.
>
> A: We will use "heterogeneous" to describe the difference between observation and action and "multi-modal" for the action distribution in revision. We will also include an experiment to clarify the multi-modality.
>
> The use of $l_1$ norm objective in Eq. (3) follows UniDB. We will make the explanation more explicit and add the missing norm-ablation results in the revision. Training with $l_1$ norm is slightly better than $l_2$ norm. The average success rate using $l_1$ norm on the Adroit benchmark is 0.81, but $l_1$ norm can also achieve 0.80.
> > W2: Possible misunderstanding of "one-to-one".
>
> A: We will remove the "one-to-one" in revision. The phrase “one-to-one” refers to that diffusion bridge connects one distribution to another not to a deterministic observation-to-action mapping for each sample.
> > W3.1 & Q1: Missing hyperparameter on simulation tasks and clarification on $\alpha$.
>
> A: For fair comparison, we use the same common hyperparameters as prior work (DP3, FlowPolicy) across all simulation tasks, and set $\alpha=1$ in BridgePolicy for all tasks without per-task tuning.
> > W3.2: Repetitive emphasis of the general idea.
>
> A: We will streamline these parts to reduce repetition and improve clarity.
> > W3.3: Hypothesis in Theorem 3.1.
>
> A: We will add the hypothesis that the sampling noise is the same for both predictions and the definition of $\epsilon$ in revision.
> > W3.4: Oversold of the "fusion" component.
>
> A: We will soften the related claims and revise to present the fusion component more precisely, as a pipeline of point cloud downsampling + MLP encoding + cross-attention.
> > W3.5 & W3.6: Suggestions for the content organization and formatting.
>
> A: We will move the corresponding equations to the Appendix, fix the incorrect table reference, reorder the relevant tables and figures to make the presentation clearer.
> > W3.7: Evaluation Protocol.
>
> A: The evaluation of BridgePolicy directly follows the standard protocol used in prior generative policy work for fair comparison (like DP3 and FlowPolicy) It is applied uniformly across methods, without any specific settings favoring BridgePolicy.
> > Q3: Real-world ablation study.
>
> Thanks for the suggestion. We additonally **conduct a real-world experiment.** Specifically, we use 10, 20 and 50 demonstrations to train BridgePolicy and FlowPolicy on a new real-world pick-place task. The success rates are 0.5, 0.7, 0.9 for BridgePolicy and 0.4, 0.6 and 0.9 for FlowPolicy, showing the consistent trend in simulation.

---

> > ### Author Rebuttal · Reviewer_z5Nz · 2026-04-01
> >
> > The rebuttal answer covered four different aspects:
> > - Some explanations for the differences from prior work (W1.3 & Q4), I may not be expert enough to debate the expressiveness of SDEs compared to ODEs in a robotic setup. I still view these approaches as structurally close, but so long as the claims are softened, I think it's ok.
> > - Promises that the final version will be smoothed, softened, and more streamlined. Honestly, on this, there is not much more the authors can do given the shortness of the rebuttal (and the fact that they should not provide a revised version). I thank the authors for having considered my feedback on the phrasing/presentation, but unlike the other aspects of the rebuttal, these promises can hardly make the score increase (this has more to do with how ICML works).
> > - I appreciate the technical precision on the stop-gradient and the response to W3.1/Q1.
> > - Finally, the two new experiments: the l1 norm and the ablation study on the number of demos. I would like some precision: (hence the (b) "Partially resolved")
> > 1) About the l1 normalization, there seem to be a typo in the sentence, "The average success rate using $l1$ norm on the Adroit benchmark is 0.81, but $l1$ norm can also achieve 0.80." One $l1$ should be $l2$ I guess.
> > 2) About the new real-world ablation study on the number of demonstrations. **In the original paper, on the Pick-Place task, BridgePolicy has a success rate of 0.8, compared to 0.5 for FlowPolicy, using 50 demonstrations** (Tab 2). In the new ablation study at the end of the rebuttal, the authors claim:
> > > Specifically, we use 10, 20 and 50 demonstrations to train BridgePolicy and FlowPolicy on a new real-world pick-place task. The success rates are 0.5, 0.7, 0.9 for BridgePolicy and 0.4, 0.6 and 0.9 for FlowPolicy
> >
> > These new results seem very inconsistent with the ones reported in the paper. Especially for 50 demonstrations, BridgePolicy changed from 0.8 to 0.9; and FlowPolicy from 0.5 to 0.9. Did the task change? Even if it did change, if, depending on the exact setup, the results can vary this much, then should the new results be averaged with the old ones?
> >
> > For now, I would like to keep my rating as a weak accept.

---

> > > ### Author Response · Authors · 2026-04-02
> > >
> > > Thank you for the thoughtful follow-up.
> > >
> > > > Some explanations for the differences from prior work (W1.3 & Q4), I may not be expert enough to debate the expressiveness of SDEs compared to ODEs in a robotic setup. I still view these approaches as structurally close, but so long as the claims are softened, I think it's ok.
> > >
> > > Thank you for your understanding and for acknowledging our clarification on prior work.
> > >
> > > > Promises that the final version will be smoothed, softened, and more streamlined. Honestly, on this, there is not much more the authors can do given the shortness of the rebuttal (and the fact that they should not provide a revised version). I thank the authors for having considered my feedback on the phrasing/presentation, but unlike the other aspects of the rebuttal, these promises can hardly make the score increase (this has more to do with how ICML works).
> > >
> > > Thank you for your understanding. We value your advice on the phrasing and presentation, which really helps to improve the quality of our paper. We commit to making the corresponding improvement in the revision.
> > >
> > > > I appreciate the technical precision on the stop-gradient and the response to W3.1/Q1.
> > >
> > > Thank you for the acknowledgement of our clarification on the stop-gradient and provided training details.
> > >
> > > > Finally, the two new experiments: the $l_1$ norm and the ablation study on the number of demos. I would like some precision: (hence the (b) "Partially resolved")
> > >
> > > > Q1: About the $l_1$ normalization, there seems to be a typo in the sentence, "The average success rate using $l_1$ norm on the Adroit benchmark is 0.81, but $l_2$ norm can also achieve 0.80." One $l_1$ should be $l_2$ I guess.
> > >
> > > A: You are right, it is indeed that "Training with $l_1$ norm is slightly better than $l_2$ norm. The average success rate using $l_1$ norm on the Adroit benchmark is 0.81, but $l_2$ norm can also achieve 0.80." Thank you for pointing out this typo.
> > >
> > > > Q2: These new results seem very inconsistent with the ones reported in the paper. Especially for 50 demonstrations, BridgePolicy changed from 0.8 to 0.9; and FlowPolicy from 0.5 to 0.9. Did the task change?
> > >
> > > A: Regarding the real-world ablation on the number of demonstrations, due to the space limit of the rebuttal, we were unable to provide sufficient experimental details, so we would like to clarify them here. This ablation was conducted on a new real-world **pick-place-cup** setup rather than the original **pick-place-bowl** in Table 2. The experimental setting is shown in https://anonymous.4open.science/r/tmp-844F/Fig.png.
> > >
> > > Concretely, the new task requires placing an upside-down cup into a basket. On the one hand, the new setting is somewhat easier than the original one since the cup and the basket are **larger (deeper)** than the previous bowl and basket, which makes the manipulated object and the target location **more clearly observable in the 3D point cloud**.
> > >
> > > On the other hand, as also reflected by the results reported in Table 2, FlowPolicy can **vary considerably** across different real-world setups while BridgePolicy **remains relatively stable**.
> > >
> > > > Even if it did change, if, depending on the exact setup, the results can vary this much, then should the new results be averaged with the old ones?
> > >
> > > Considering these differences are substantial, we believe it **would not be appropriate to directly average** the new ablation results with the original results in Table 2, but we will add this new ablation and use **pick-place-bowl** and **pick-place-cup** to distinguish it from the original results in the revision.

---

### Decision · Program_Chairs · 2026-04-30

**Decision:**

Accept (regular)

**Comment:**

Generally, reviewers found this paper technically sound, well-written, and somewhat novel. It may be useful to at least some fraction of the ICML community. Reviewers emphasized the importance of clarifying the novelty claims with accurate (not overstated) claims.